# Retraining-Free Merging of Sparse Mixture-of-Experts via Hierarchical Clustering

## Abstract

Sparse Mixture-of-Experts (SMoE) models represent a significant breakthrough in large language model development. These models enable performance improvements without a proportional increase in inference costs. By selectively activating a small set of parameters during task execution, SMoEs enhance model capacity. However, their deployment remains challenging due to the substantial memory footprint required to accommodate the growing number of experts. This constraint renders them less feasible in environments with limited hardware resources. To address this challenge, we propose Hierarchical Clustering for Sparsely activated Mixture of Experts (HC-SMoE), a task-agnostic expert merging framework that reduces SMoE model parameters without retraining. Unlike previous methods, HC-SMoE employs hierarchical clustering based on expert outputs. This approach ensures that the merging process remains unaffected by routing decisions. The output-based clustering strategy captures functional similarities between experts, offering an adaptable solution for models with numerous experts. We validate our approach through extensive experiments on eight zero-shot language tasks and demonstrate its effectiveness in large-scale SMoE models such as Qwen and Mixtral. Our comprehensive results demonstrate that HC-SMoE consistently achieves strong performance, which highlights its potential for real-world deployment.

## 1 Introduction

The exponential growth in model parameters for Transformer-based architectures in natural language processing (NLP) has led to significant performance improvements across various tasks (Chowdhery et al., 2022; OpenAI et al., 2024; Team et al., 2024). Nevertheless, this increase in size has resulted in challenges for real-world deployment and accessibility due to heightened inference latency and computational requirements (Bommasani et al., 2022) Sparsely activated Mixture of Experts (SMoE) models have emerged as a promising solution to this challenge. SMoE architectures employ a sparse activation mechanism, wherein only a subset of the model's parameters, or '*experts*', are activated for each input token. This design enables extensive parametric capacity without a proportional increase in computational cost during inference, as shown in previous works (Shazeer et al., 2017; Fedus et al., 2022). However, despite improvements in inference latency, the overall size of SMoE architectures poses a significant challenge for memory usage, and the efficient reduction of SMoE model size during inference has become a critical area of concern. Recently, the authors in Liu et al. (2023) have identified high representational similarity among experts and suggested a solution to enhance expert diversity within an SMoE. Such an observation is further substantiated by the empirical results provided in Lu et al. (2024). The collective evidence from these previous studies therefore suggests that model parameters in contemporary SMoE architectures may be redundant, which points to avenues for potential optimization and efficiency improvements.

To address the challenges associated with redundant parameters in SMoE models, researchers have proposed various approaches in the past few years. Early endeavors focused on task-specific expert pruning (Chen et al., 2022), which progressively eliminates non-essential experts and ultimately results in a single-expert dense model tailored for a specific downstream task. However, such approaches often necessitate extensive fine-tuning to mitigate performance degradation caused by pruning. Inspired by this limitation, several studies (Lu et al., 2024; He et al., 2024) have explored retraining-free expert pruning methods. For instance, Lu et al. (2024) proposed directly trimming experts by using the smallest output loss relative to the original model as an indicator of importance,

without any fine-tuning. On the other hand, He et al. (2024) introduced a more scalable pruning process based on routing scores. The other line of studies (Li et al., 2024) suggested replacing pruning approaches with a merging method to more effectively leverage shared knowledge among experts and consolidate information from the most significant ones. Nonetheless, according to our experimental investigation in 4, this method exhibits inferior generalizability in task-agnostic setups.

In light of the aforementioned challenges, this paper proposes a *retraining-free*, *scalable*, and *task-agnostic* framework, termed **H**ierarchical **C**lustering for **S**parsely Activated **M**ixture **o**f **E**xperts (**HC-SMoE**), that concentrates on reducing the parameters of an SMoE model through hierarchical clustering based on expert outputs and frequency-weighted merging. The hierarchical clustering methodology offers two key advantages. First, instead of using the grouping approach adopted in Li et al. (2024), which only calculates the similarity among the experts once, hierarchical clustering can better maintain inter-cluster diversity and intra-cluster similarity by iterative comparison. Second, in contrast to the similarity metric used in Li et al. (2024), HC-SMoE leverages experts' outputs rather than router logits, enhancing generalizability against dataset-specific information. This observation is supported by the experimental results presented in Section 4. To evaluate our design in a task-agnostic experimental setup, we first perform clustering and merging based on the C4 dataset (Raffel et al., 2020), and then assess accuracy across eight zero-shot language tasks (Lu

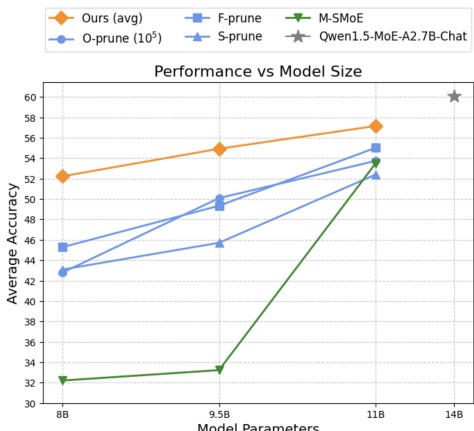

Figure 1: A comparison of the average accuracy across eight LM-Harness benchmarks on Qwen1.5-MoE-A2.7B-Chat (Team, 2024). HC-SMoE (Ours) outperforms existing retraining-free pruning and merging baselines, achieving superior results while removing 25%, 37.5%, and 50% of SMoE expert parameters.

et al., 2024). As shown in Fig. 1, our method achieves performance comparable to the original model (i.e., Qwen) and outperforms the best-performing baseline by 6.95% and 2.14% under the 8B and 11B parameters setups, respectively. Moreover, our results in Section 4 further demonstrate that HC-SMoE outperforms all baselines with the Mixtral 8x7B setup. Our contributions are fourfold:

- To the best of our knowledge, this study presents the first retraining-free, task-agnostic SMoE merging strategy that scales efficiently with the number of experts.
- We demonstrate the effectiveness of using expert outputs as the similarity metric for clustering, as compared to router logits or weights employed by prior arts.
- We highlight the importance of clustering quality prior to merging and illustrate that the proposed hierarchical clustering provides robust and reliable results for expert grouping.
- Our experimental results reveal that HC-SMoE consistently exhibits superior performance across various benchmarks and demonstrates its efficacy on large-scale SMoE models.

## 2 BACKGROUND AND RELATED WORKS

This section presents the essential background and related works. Section 2.1 introduces the SMoE architecture, while Section 2.2 discusses various pruning and merging methods based on SMoE.

### 2.1 SPARSELY ACTIVATED MIXTURE-OF-EXPERTS (SMOE)

The SMoE model comprises multiple SMoE layers, each of which contains a set of expert neural networks and a router network. Consider an input token $x$, a set of expert neural networks $\{E_1, E_2, ..., E_n\}$, and a router network $R$. The output $y$ of an SMoE layer is computed as a weighted sum of the expert network outputs, which can be expressed as Eq. (1):

$$y = \sum_{i=1}^{n} P_i(x) \cdot E_i(x), \quad (1) \qquad\qquad E(x) = (\sigma(xW_{\text{gate}}) \odot (xW_{\text{up}}))W_{\text{down}}, \qquad (2)$$

Table 1: A Comparison of different approaches for reducing the number of experts in SMoE. Please note that F-prune is detailed in Section 4.

| Method | Retraining-free | Task-agnostic | Scalable | Strategy |
|---|---|---|---|---|
| TSEP (Chen et al., 2022) | ✗ | ✗ | ✓ | Pruning |
| O-prune (Lu et al., 2024) | ✓ | ✓ | ✗ | Pruning |
| S-prune (He et al., 2024) | ✓ | ✓ | ✓ | Pruning |
| F-prune | ✓ | ✓ | ✓ | Pruning |
| M-SMoE (Li et al., 2024) | ✗ | ✗ | ✓ | Merging |
| HC-SMoE (Ours) | ✓ | ✓ | ✓ | Merging |

where $P_i(x)$ represents the routing score from $R$ for the $i^{th}$ expert, and $E_i(x)$ denotes the output of the $i^{th}$ expert network. Building upon this formulation, the experts in both Qwen (Team, 2024) and Mixtral (Jiang et al., 2024) adopt the structure of LLaMA (Touvron et al., 2023). Specifically, the feed-forward network (FFN) within each expert consists of three linear layers that function as Eq. (2), where $\odot$ signifies element-wise multiplication, $W_{\text{up}}, W_{\text{gate}} \in \mathbb{R}^{d_h \times d_m}$, and $W_{\text{down}} \in \mathbb{R}^{d_m \times d_h}$ denote the weight matrices, and $\sigma$ is the activation function, specifically Sigmoid Linear Unit (SiLU, also known as the swish function) (Elfwing et al., 2018). In practice, an efficient implementation of the routing function utilizes a top-$k$ routing strategy (Shazeer et al., 2017; Fedus et al., 2022), which selects only the top $k$ experts based on the highest logits from a linear transformation of the input. The softmax operation is then applied to the $k$ largest logits, which result in a sparse activation of experts. This procedure reduces the computational overhead by activating only a small subset of the available experts. Mathematically, this procedure can be expressed as follows:

$$P(x) = \text{softmax}(\text{topK}(R(x))) = \text{softmax}(\text{topK}(xW_R)), \tag{3}$$

where $R(x)$ represents the routing-logits and $W_R$ denote a learnable parameter matrix. Such a sparsely activated MoE stucture leverage this mechanism to scale efficiently while maintaining its performance. By focusing computation on the most relevant experts for each input token, the SMoE model is able to achieve a balance between the computational efficiency and the task performance.

## 2.2 EXPERT PRUNING AND MERGING

This section discusses prior methods for reducing the number of experts in an SMoE. Table 1 offers a summary of them. We begin with the pruning strategies, and then examines the merging techniques.

Pruning strategies for SMoE models have been the focus of several recent studies. Chen et al. (2022) introduced *Task-Specific Expert Pruning (TSEP)*, which iteratively fine-tunes the model while pruning experts to gradually reduce the number of active experts for a specific downstream task. Despite its effectiveness, the extensive fine-tuning required renders this process time-consuming and computationally expensive, limiting its practicality for large-scale models. Lu et al. (2024) proposed a method which we refer to as *O-prune* in this study, which directly prunes experts using a combinatorial search to minimize the *Output* loss in a retraining-free, task-agnostic, zero-shot setting. This approach begins by identifying the number of experts to retain in each layer. It then conducts a layer-wise search across all possible expert combinations, selecting the one that minimizes output loss compared to the original model. Nevertheless, this method directly prunes experts deemed unimportant, forfeiting the opportunity to leverage their knowledge. Furthermore, the exhaustive combinatorial search becomes computationally infeasible for models with a large number of experts. For instance, reducing 50% of the experts in a model like Qwen, which has 60 experts, necessitates enumerating approximately $C(60, 30) \approx 10^{18}$ combinations per layer, rendering it impractical for large-scale applications. He et al. (2024) introduced an efficient expert trimming technique, denoted as *S-prune*, due to its reliance on the router *Score*. In this approach, each expert's router-score $P(x)$ is accumulated globally, and only the top-scoring experts are retained, while others are pruned. This strategy offers more flexibility than that in Lu et al. (2024), as it enables different layers to retain varying numbers of experts.

Model merging techniques have emerged as a promising approach to combine the strengths of multiple models. ZipIt (Stoica et al., 2024) introduces a model merging technique that allows models with the same architecture but trained on different tasks to be merged without retraining. It utilizes pairwise feature correlation to merge features both within a single model and across different

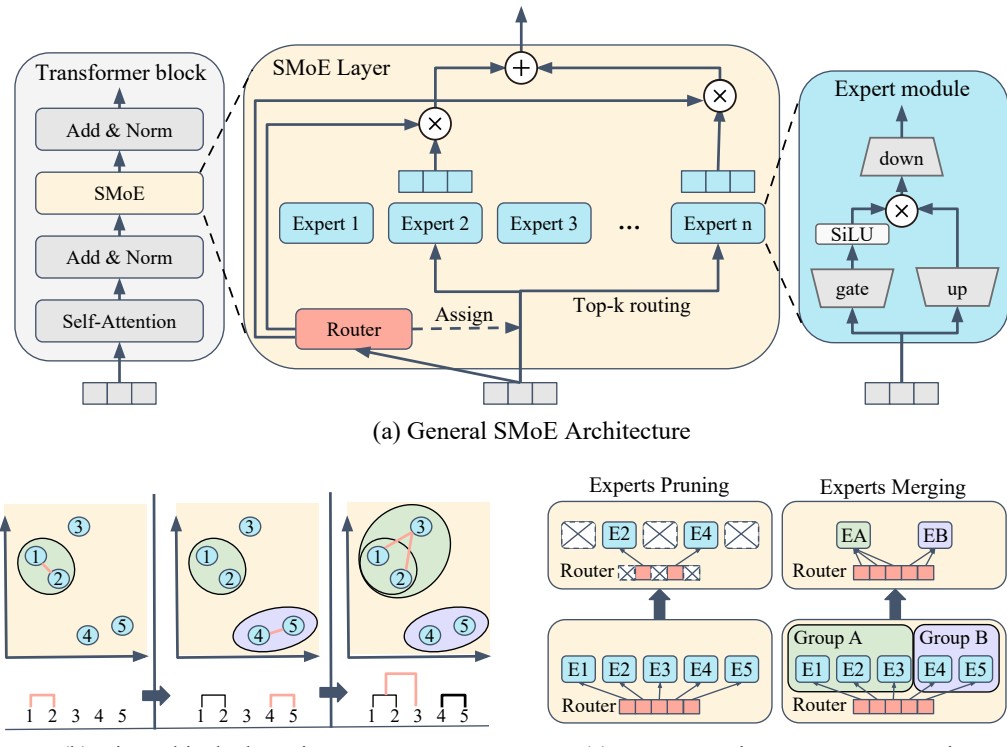

(a) General SMoE Architecture

(b) Hierarchical Clustering on Experts

(c) Expert Pruning v.s. Expert Merging

Figure 2: (a) The general SMoE architecture. (b) Illustration of the proposed hierarchical clustering strategy based on expert outputs. (c) Comparison of expert pruning and expert merging strategies.

models, offering flexibility in choosing correlated features. Since expert merging can be considered a multi-model merging problem, we extend ZipIt to this context. However, its extensive feature correlation computation makes it time-consuming and less effective for expert merging scenarios. M-SMoE (Li et al., 2024) proposes a three-step pipeline for expert merging in SMoE models. It first selects dominant experts based on activation frequency to decide which experts to retain in each layer, then uses router logits $R(x)$ to group experts, followed by frequency-based merging. However, in task-agnostic settings without retraining, relying on frequency information for clustering proves ineffective. This approach faces two main issues. First, frequency varies across tasks, as shown in Appendix C, making it an unreliable indicator for deciding how many experts to retain in each layer. Second, high-frequency experts within the same layer are rarely merged, overlooking their functional similarities in the feature space. Additionally, grouping based on router information is problematic, as it depends on dataset-dependent statistics. Together, the limitations can potentially hinder the model's ability to maintain performance over diverse tasks without access to task data.

## 3 METHODOLOGY

In this section, we introduce the proposed HC-SMoE. Section 3.1 first provides a formal problem definition. Section 3.2 then outlines the merging pipeline. Sections 3.2.1, 3.2.2, and 3.2.3 further explains the similarity metric, clustering method, and merging criterion of HC-SMoE, respectively.

### 3.1 PROBLEM DEFINITION

In this study, we address the challenge of reducing the space complexity of an SMoE model through a process termed *expert merging*. This process consolidates existing experts in an SMoE layer into a smaller set while preserving the model's performance. Each SMoE layer initially contains $n$ experts, as defined in Section 2.1. We aim to merge these experts into $r$ clusters, where $r$ represents the target number of experts after merging. For the $i$-th cluster, denoted as $C_i = \{E_0^i, E_1^i, \ldots, E_{|C_i|}^i\}$, $|C_i|$ represents the number of original experts assigned to this cluster. During merging, all experts within a cluster combine into a single new expert, which effectively reduces the total number of experts to $r$.

The distribution of original experts across all clusters satisfies $\sum_{i=1}^{r} |C_i| = n$, which ensures that all original experts are accounted for in the merging process. To achieve this objective, we explore two grouping strategies. Static grouping maintains exactly $r$ experts in each layer after merging, while dynamic grouping allows this number to vary per layer while maintaining an average of $r$. Our proposed HC-SMoE adopts static grouping, similar to *O-prune*, while *F-prune* and *S-prune* employ dynamic grouping. Throughout this process, the router network $R$ remains unchanged. If an input token was previously assigned to any expert within a merged group, it is routed to the corresponding merged expert. This approach preserves input dimensionality while reducing the number of experts.

The expert merging problem presents several unique challenges. Unlike conventional model merging methodologies, where the entities to be merged are predetermined, expert merging requires an initial clustering of experts into groups before the merging process can be executed. This additional clustering step substantially increases the complexity of the problem, as it necessitates determining which subsets of experts can be grouped in a manner that minimizes performance loss in the model. Since training an MoE model demands substantial GPU memory, we address this problem without retraining and utilize a non-benchmark dataset to collect information for the expert merging process.

## 3.2 HIERARCHICAL CLUSTERING FOR SPARSELY ACTIVATED MIXTURE OF EXPERTS

In this section, we elaborate on HC-SMoE, a methodology designed with the philosophy of being retraining-free, task-agnostic, and scalable. HC-SMoE employs a two-stage pipeline comprising clustering and merging, which are illustrated in Fig. 2 (b) and the right-hand side of Fig. 2 (c), respectively. The first clustering stage determines the optimal grouping of experts for subsequent merging. Our approach demonstrates that utilizing averaged expert outputs as the similarity metric for clustering yields superior effectiveness compared to methods relying on averaged router logits or weights. Furthermore, we emphasize the critical importance of clustering quality and illustrate that hierarchical clustering with average linkage produces robust model quality post-merging. The second stage encompasses the merging of experts within each identified cluster. Our experimental findings indicate that given effective clustering, the choice of merging method (e.g., average, frequency, or alternative merging approaches) consistently results in strong performance across all benchmarks. The above components are further explained in Subsections 3.2.1, 3.2.2, and 3.2.3.

### 3.2.1 AVERAGED EXPERT OUTPUTS AS SIMILARITY METRIC

Establishing a robust and reliable expert similarity metric is crucial for HC-SMoE prior to performing clustering. HC-SMoE employs an average output strategy, which proves advantageous due to its effectiveness in capturing expert functionality. The method utilizes averaged expert outputs $\frac{1}{T} \sum_{i=1}^{T} E(x_i)$ over a calibration dataset with $T$ tokens to effectively capture each expert's functional behavior. The expert feature vectors represent the final transformed data, encapsulating both the input's contextual information and the expert's learned transformations. We evaluated multiple similarity metrics for comparing expert behavior, including router logits $R(x)$ (Li et al., 2024) and expert weights, represented as flatten($W_{\text{gate}}||W_{\text{down}}||W_{\text{up}}$), where $||$ denotes concatenation. Our empirical study, detailed in Section 4.3, demonstrates that expert outputs provide a more effective similarity metric. The following analysis discusses the benefits of expert outputs over these metrics.

Router logits reflect assignment preferences influenced by local factors such as specific input distributions rather than the actual functional similarity between experts. The routing patterns can vary significantly across tasks and thus limit their utility as task-agnostic similarity metrics. Although expert weights contain rich parameter information, their processing is computationally expensive. Furthermore, a prior study (Liu et al., 2023) has shown that SMoE experts often exhibit high similarity in the parameter space, which reduces their effectiveness as a distinct measure of expert functionality. In contrast, averaged expert outputs provide a direct representation of each expert's functional behavior and improve the identification of functionally similar experts. Previous research (Li et al., 2016; Stoica et al., 2024) indicates that models with similar outputs are likely to perform analogous functions. This correlation validates the efficacy of clustering based on output similarity to preserve model performance after merging and offers the foundation for HC-SMoE's output merging strategy.

### 3.2.2 HIERARCHICAL CLUSTERING

With a reliable expert similarity metric established, the subsequent step involves selecting an appropriate clustering method. Unlike conventional model merging, which combines predefined element groups, expert merging necessitates a more flexible approach to group experts before merging. Although several potential strategies exist, as discussed in Section 4.3, our analysis reveals that hierarchical clustering (Patel et al., 2015), as illustrated in Fig. 2 (b), is well-suited for this task. This method exhibits optimal characteristics for expert grouping. Hierarchical clustering initiates by considering each expert as an individual entity and progressively merges them based on their cluster distance. This process facilitates adaptive grouping throughout the procedure. The approach continuously recalculates distances between clusters, which ensures the grouping of functionally similar experts. As a result, hierarchical clustering is able to outperform the K-means and one-shot grouping (Li et al., 2024). Section 4.3 provides an ablation study for different clustering strategies. In this study, Euclidean distance is employed to measure the distance between two experts, expressed as:

$$d(e_i, e_j) = ||e_i - e_j||_2 \tag{4}$$

where $e_i$ and $e_j$ denote the metric values for computing distances between experts $i$ and $j$. Our investigation includes three linkage methods in hierarchical clustering: *single*, *complete*, and *average*.

$$\text{single:} \quad \min_{a \in A, b \in B} d(a, b) \tag{5}$$

$$\text{complete:} \quad \max_{a \in A, b \in B} d(a, b) \tag{6}$$

$$\text{average:} \quad \frac{1}{|A| \cdot |B|} \sum_{a \in A} \sum_{b \in B} d(a, b) \tag{7}$$

where $A$ and $B$ represent clusters, and $a$ and $b$ denote experts that belong to these clusters. Single linkage defines cluster distances through the closest pair of elements, while complete linkage uses the maximum distance and often produces overly compact clusters that miss subtle similarities. Average linkage considers the mean pairwise distance between cluster elements and achieves an optimal balance. As demonstrated in Table 4, both single and average linkage produce satisfactory results, but average linkage achieves consistently superior performance. As a result, the proposed HC-SMoE adopts the average linkage method by default to ensure high intra-cluster similarity and low inter-cluster similarity. This balance effectively preserves expert characteristics during merging.

### 3.2.3 EXPERT MERGING

Upon completion of clustering, HC-SMoE proceeds to merge the experts within each cluster, as illustrated in Fig. 2 (c). Our empirical evidence indicates that while the choice of merging method does influence the overall performance, its impact is relatively modest compared to the significance of clustering results. Specifically, HC-SMoE merges the clustered experts on the weight space as:

$$\hat{E}_i = \sum_{j=1}^{|C_i|} \alpha_j E_j, \quad \sum_{j=1}^{|C_i|} \alpha_j = 1 \tag{8}$$

where parameter $\alpha_j$ denotes weight of merging expert $j$. This study considers three different merging strategies: *average* merging, *frequency-weighted* merging, as well as *fixed-dominant* merging. In average merging, $\alpha_j = \frac{1}{|C_i|}$. In frequency-weighted merging, $\alpha_j$ denotes the usage frequency of expert $j$. On the other hand, fixed-dominant merging, a methodology introduced in this study, represents an efficient adaptation of ZipIt specifically developed for merging experts in SMoE models.

In fixed-dominant merging, the expert closest to the cluster center is selected as the dominant expert, which serves as a reference point. The features of the other experts in the cluster are then re-ordered based on their correlation with the dominant expert's features. Subsequently, an averaging operation is performed between the dominant expert and the re-ordered features of the remaining experts. This method is specifically tailored for SMoEs, as it preserves the structural characteristics of the dominant expert while allowing efficient merging. Unlike the original ZipIt algorithm, which requires exhaustive pairwise comparisons, our adaptation is computationally efficient and ensures the preservation of key characteristics of the dominant expert, making it highly suitable for the

SMoE framework. An ablation study for different merging approaches is presented in Section 4.3. For elaboration on fixed-dominant merging and comparison to ZipIt, please refer to Appendix A.2.

# 4    EXPERIMENTAL RESULTS

In the following sections, we first describe the experimental setups in Section 4.1. Next, we compare the performance of HC-SMoE against various SMoE model pruning and merging baselines. Finally, we present an ablation study to assess the effectiveness of each component in HC-SMoE's design.

## 4.1    EXPERIMENTAL SETTINGS

We conduct experiments on two SMoE models: Qwen1.5-MoE-A2.7B (henceforth Qwen) (Team, 2024) and Mixtral 8x7B (Jiang et al., 2024). For Qwen, we explore two levels of reduction: merging the number of experts from 60 to 45 and further to 30 per layer. This corresponds to a reduction in parameters from 14.3B to 11.2B (denoted as Qwen 45x2.7B), and subsequently to 8.1B (denoted as Qwen 30x2.7B). Similarly, Mixtral 8x7B undergoes reduction from eight to six experts and then to four experts per layer, decreasing the total parameters from 46.7B to 35.6B (denoted as Mixtral 6x7B) and further to 24.3B (denoted as Mixtral 4x7B). This graduated approach enables the evaluation of expert merging impact at different levels of model reduction. Experiments on Mixtral 8x7B and Qwen are conducted on eight NVIDIA A100 GPUs and four NVIDIA V100 GPUs, respectively.

To evaluate our method in a task-agnostic setting, we utilize eight tasks using the EleutherAI Language Model Evaluation Harness (Gao et al., 2024). These are designed to cover various aspects of language understanding and reasoning, including both Challenge and Easy sets in AI2 Reasoning Challenge (ARC) (Clark et al., 2018), BoolQ (Clark et al., 2019), HellaSwag (Zellers et al., 2019), Massive Multitask Language Understanding (MMLU) (Hendrycks et al., 2021), OpenBookQA (Mihaylov et al., 2018), Recognizing Textual Entailment (RTE) (Bentivogli et al., 2009) and Winograd Schema Challenge (Sakaguchi et al., 2021). We report zero-shot accuracy on those benchmarks.

For our comparisons, three pruning baselines are employed: *O-prune* (Lu et al., 2024), *S-prune* (He et al., 2024), and *F-prune*. *F-prune*, where 'F' denotes frequency, adheres to the same methodology as *S-prune*. However, it employs frequency as the criterion for pruning experts, in contrast to *S-prune* which utilizes router logits. Due to the high computational complexity of *O-prune* on Qwen, a random sampling of $10,000$ possible expert sets in each layer is performed instead. The set with the smallest output difference from the original model is selected, denoted as *O-prune ($10^5$)* in the Qwen experiments. In addition, M-SMoE is included as the merging baseline and applied in a task-agnostic setting without retraining to ensure a fair comparison. All baselines and HC-SMoE require a calibration dataset to estimate input statistics. This dataset is constructed by sampling from the C4 corpus (Raffel et al., 2020), concatenating extracted text into 32 sequences of $2,048$ tokens each.

## 4.2    PERFORMANCE COMPARISONS

This section presents a comprehensive comparison of the performance of the models reduced by the proposed HC-SMoE against the original SMoE models and the baselines. The analysis encompasses various model sizes and tasks, and provides insights into the efficacy and scalability of the proposed HC-SmoE method. As presented in Tables 2 and 3, the M-SMoE baseline exhibits the lowest performance across all benchmarks, indicating the ineffectiveness of router-logit-based grouping in a task-agnostic setting. *O-prune* demonstrates suboptimal performance, particularly on Qwen, due to its limitations in evaluating all possible expert combinations. This results in a substantial performance decline compared to Mixtral. In contrast, HC-SMoE demonstrates consistent superiority over these baselines, irrespective of model size, and proves applicable to different number of experts.

It is noteworthy that Qwen 45x2.7B and Mixtral 4x7B achieve comparable scores despite a twofold difference in parameter count. This observation substantiates the scalability of HC-SMoE to SMoE models with a higher number of experts. With a 25% reduction in experts, our method even surpasses the original model on certain tasks, such as Mixtral 6x7B on BoolQ and Qwen 45x2.7B on RTE. This improvement can be attributed to the reduction of expert redundancy after merging. In this configuration, both Qwen and Mixtral exhibit an average performance gap of less than 3% compared to their original models. Even with a 50% reduction, HC-SMoE applied to Qwen maintains a gap of

Table 2: Zero-shot performance evaluation of different expert pruning and merging methods on Qwen with reducing experts to 45 and 30 per layer. **HC-SMoE (avg)** stands for average linkage method when performing hierarchical clustering. **HC-SMoE (single)** stands for single linkage.

| Model | Method | ARC-c | ARC-e | BoolQ | HellaSwag | MMLU | OBQA | RTE | Winogrande | Average |
|---|---|---|---|---|---|---|---|---|---|---|
| Qwen 60x2.7B | None | 0.3951 | 0.7012 | 0.8135 | 0.5932 | 0.6047 | 0.310 | 0.7329 | 0.6559 | 0.6008 |
| Qwen 45x2.7B | O-prune ($10^5$) | 0.3268 | 0.6111 | 0.7566 | 0.5388 | 0.5150 | 0.268 | 0.6498 | 0.6330 | 0.5374 |
|  | F-prune | 0.3490 | 0.5989 | 0.7618 | 0.5441 | 0.4560 | **0.282** | **0.7690** | 0.6409 | 0.5502 |
|  | S-prune | 0.3464 | 0.6061 | 0.7128 | 0.5228 | 0.4930 | 0.264 | 0.6534 | 0.5935 | 0.5240 |
|  | M-SMoE | 0.3473 | 0.6157 | 0.7544 | 0.5157 | 0.4182 | 0.262 | 0.7292 | 0.6377 | 0.5350 |
|  | HC-SMoE (avg) | **0.3660** | **0.6578** | **0.7948** | 0.5520 | 0.5332 | 0.272 | 0.7509 | 0.6464 | **0.5716** |
|  | HC-SMoE (single) | 0.3592 | **0.6578** | 0.7942 | **0.5578** | **0.5360** | 0.270 | 0.7292 | **0.6472** | 0.5689 |
| Qwen 30x2.7B | O-prune ($10^5$) | 0.2568 | 0.4449 | 0.6496 | 0.4351 | 0.2907 | 0.202 | 0.6065 | 0.5375 | 0.4279 |
|  | F-prune | 0.2765 | 0.4718 | 0.6587 | 0.4330 | 0.3023 | 0.230 | 0.6570 | 0.5927 | 0.4528 |
|  | S-prune | 0.2500 | 0.4756 | 0.6388 | 0.4041 | 0.3471 | 0.196 | 0.6209 | 0.5146 | 0.4309 |
|  | M-SMoE | 0.1945 | 0.2786 | 0.4462 | 0.2837 | 0.2475 | 0.160 | 0.4477 | 0.5185 | 0.3221 |
|  | HC-SMoE (avg) | **0.3532** | **0.6149** | 0.7535 | **0.4695** | 0.4534 | **0.228** | **0.6606** | **0.6456** | 0.5223 |
|  | HC-SMoE (single) | 0.3524 | 0.6153 | **0.7661** | 0.4661 | **0.4537** | **0.228** | 0.6534 | 0.6306 | 0.5207 |

Table 3: Zero-shot performance evaluation of different expert pruning and merging methods on Mixtral 8x7B with reducing experts to six and four per layer.

| Model | Method | ARC-c | ARC-e | BoolQ | HellaSwag | MMLU | OBQA | RTE | Winogrande | Average |
|---|---|---|---|---|---|---|---|---|---|---|
| Mixtral 8x7B | None | 0.5648 | 0.8422 | 0.8505 | 0.6490 | 0.6712 | 0.350 | 0.7112 | 0.7593 | 0.6748 |
| Mixtral 6x7B | O-prune | **0.5205** | 0.8009 | 0.8352 | 0.6115 | 0.5741 | 0.316 | 0.6606 | **0.7719** | 0.6363 |
|  | F-prune | 0.5009 | 0.7904 | 0.7725 | 0.5990 | 0.5099 | 0.326 | 0.5596 | 0.7672 | 0.6032 |
|  | S-prune | 0.4991 | 0.7891 | 0.7801 | 0.5984 | 0.5103 | **0.340** | 0.5704 | 0.7735 | 0.6076 |
|  | M-SMoE | 0.2619 | 0.5564 | 0.5208 | 0.4320 | 0.2503 | 0.194 | 0.5271 | 0.5848 | 0.4159 |
|  | HC-SMoE (avg) | 0.5145 | 0.8043 | **0.8554** | 0.6142 | 0.6043 | 0.324 | **0.6715** | 0.7514 | **0.6425** |
|  | HC-SMoE (single) | 0.5154 | **0.8123** | **0.8554** | **0.6163** | **0.6053** | 0.310 | **0.6715** | 0.7403 | 0.6408 |
| Mixtral 4x7B | O-prune | 0.4394 | 0.7327 | 0.8046 | 0.5660 | 0.4584 | 0.286 | 0.5668 | **0.7285** | 0.5728 |
|  | F-prune | 0.4352 | 0.7290 | 0.7520 | 0.5293 | 0.3739 | **0.290** | 0.5560 | 0.7245 | 0.5487 |
|  | S-prune | 0.2235 | 0.4339 | 0.6300 | 0.4250 | 0.2554 | 0.188 | 0.5235 | 0.5699 | 0.4062 |
|  | M-SMoE | 0.2116 | 0.2765 | 0.4954 | 0.2767 | 0.2452 | 0.108 | 0.4910 | 0.4964 | 0.3251 |
|  | HC-SMoE (avg) | 0.4573 | 0.7454 | 0.8018 | 0.5709 | 0.4571 | 0.270 | 0.5523 | **0.7285** | 0.5729 |
|  | HC-SMoE (single) | **0.4642** | **0.7483** | **0.8321** | **0.5781** | **0.4895** | 0.280 | **0.5884** | 0.7206 | **0.5877** |

merely 7.43% and outperforms the best baseline, *F-prune*, which lags behind HC-SMoE by 7.46%. These results validate the robustness and efficacy of HC-SMoE across diverse model sizes and tasks.

## 4.3 ABLATION STUDY

**Ablation on Different Linkage Methods among Different Metrics.** Table 4 presents a comparison of different linkage methods in hierarchical clustering according to various metrics: *router-logits*, *weight*, and *expert-output*. Hierarchical clustering exhibits stability due to its deterministic nature. This stability is evidenced by consistent performance across benchmarks and the highest average scores. Unlike K-means, it is not susceptible to initialization randomness, which establishes it as a more reliable clustering method. Among the different linkage methods, single linkage generally performs satisfactorily. However, average linkage emerges as the superior option and achieves the highest scores in most of the evaluated settings. The experimental results further reveal an interesting pattern in the performance of complete linkage across different metrics. When applied with the expert-output metric, complete linkage yields suboptimal results, achieving only 0.3909 on average. The performance further deteriorates with the weight metric, which reaches a mere 0.3682. On the contrary, the router-logits-based approach excels exclusively with complete linkage, and attains an average score of 0.5295. This disparity substantiates the distinctive properties of router-logits compared to weights and expert outputs. This observation can be attributed to the inherent characteristics of the similarity metrics. Router-logits align well with complete linkage since they capture the maximal boundaries between clusters. This alignment effectively reflects distinct activation patterns. In contrast, expert outputs and weights benefit from single or average linkage methods. These metrics reveal more subtle, internal similarities that may not manifest through extreme distances. Therefore, they favor linkage methods that consider average or minimal distances between cluster elements.

Table 4: Different linkage method comparisons of hierarchical clustering on Qwen 45x2.7B.

| Linkage | Metric | ARC-c | BoolQ | OBQA | RTE | Average |
|---|---|---|---|---|---|---|
| None | None | 0.3951 | 0.8135 | 0.310 | 0.7329 | 0.5629 |
| Single | *router-logits* | 0.2398 | 0.3792 | 0.180 | 0.5054 | 0.3261 |
| | *weight* | 0.3695 | 0.7676 | 0.254 | 0.7004 | 0.5229 |
| | *expert-output* | 0.3592 | 0.7942 | 0.270 | 0.7292 | 0.5382 |
| Complete | *router-logits* | 0.3677 | 0.7694 | 0.248 | 0.7329 | 0.5295 |
| | *weight* | 0.2363 | 0.4446 | 0.178 | 0.6137 | 0.3682 |
| | *expert-output* | 0.2338 | 0.6037 | 0.210 | 0.5162 | 0.3909 |
| Average | *router-logits* | 0.2073 | 0.3801 | 0.172 | 0.5018 | 0.3153 |
| | *weight* | **0.3788** | 0.7645 | 0.250 | 0.7004 | 0.5234 |
| | *expert-output* | 0.3660 | **0.7948** | **0.272** | **0.7509** | **0.5459** |

Table 5: Our HC-SMoE compares to K-means clustering. K-means-**fix** assigns the first $r$ experts as initial center. K-means-**rnd** randomly choose $r$ experts as initial center. For each task, we highlight the best performance in ==yellow==, and mark the best performance within same cluster method **bold**.

| Model | Cluster | Metric | ARC-c | BoolQ | OBQA | RTE | Average |
|---|---|---|---|---|---|---|---|
| Qwen 60x2.7B | None | None | 0.3951 | 0.8135 | 0.310 | 0.7329 | 0.5629 |
| Qwen 45x2.7B | K-means-fix | *router-logits* | 0.3669 | 0.7664 | **0.270** | 0.6679 | 0.5178 |
| | | *weight* | 0.3797 | 0.7294 | 0.268 | 0.6751 | 0.5131 |
| | | *expert-output* | **0.3925** | **0.7850** | **0.270** | **0.7184** | **0.5415** |
| | K-means-rnd | *router-logits* | **0.3797** | **0.7621** | **0.276** | 0.6029 | 0.5052 |
| | | *weight* | 0.2432 | 0.5557 | 0.154 | 0.5812 | 0.3835 |
| | | *expert-output* | **0.3797** | 0.7177 | 0.270 | **0.6968** | **0.5161** |
| | HC-SMoE | *expert-output* | 0.3646 | **0.7927** | 0.268 | **0.7449** | **0.5426** |
| Qwen 30x2.7B | K-means-fix | *router-logits* | 0.2031 | 0.4015 | 0.162 | 0.4838 | 0.3126 |
| | | *weight* | 0.2073 | **0.4960** | **0.166** | 0.509 | **0.3446** |
| | | *expert-output* | **0.2184** | 0.3786 | 0.148 | **0.5343** | 0.3198 |
| | K-means-rnd | *router-logits* | 0.2014 | 0.4168 | 0.142 | 0.5018 | 0.3155 |
| | | *weight* | 0.2108 | 0.533 | 0.174 | 0.5379 | 0.3639 |
| | | *expert-output* | **0.3370** | **0.6398** | **0.224** | **0.6065** | **0.4518** |
| | HC-SMoE | *expert-output* | **0.3515** | **0.7544** | **0.228** | **0.6631** | **0.4993** |

**K-means Clustering v.s. Hierarchical Clustering**. We next present a comparative analysis between our hierarchical clustering (HC) method and various K-means clustering strategies, underscoring the superiority of HC. Table 5 reports the performance of different initialization strategies and similarity metrics in K-means, evaluated across four benchmarks: ARC-c, BoolQ, OBQA, and RTE. These benchmarks were selected for their comprehensive coverage of language abilities, encompassing common sense reasoning, basic knowledge questions, and semantic similarity between sentence pairs. The evaluation results reveal that most post-merged models utilizing K-means experience a substantial decline in their original capabilities. For instance, even the best-performing model employing the expert-output similarity metric achieves a score 4.75% lower than our HC-SMoE results. This performance gap validates the effectiveness of our proposed HC-based method.

K-means also exhibits significant instability, particularly when juxtaposed with HC. The final performance of K-means demonstrates high sensitivity to the choice of initial cluster centers. In experiments conducted on the Qwen45x2.7B model using the weight similarity metric, we observe a substantial average accuracy reduction of 12.96% when transitioning from a fixed to a random initialization strategy. This sensitivity illuminates K-means' inherent randomness and lack of robustness. The observed instability and performance degradation in K-means clustering further accentuate the stability and efficacy of our HC-based method. These findings reinforce the superiority of HC in maintaining model performance post-merging and its resilience to initialization variability.

**Single-shot Grouping v.s. Hierarchical Clustering**. In this analysis, we follow the single-shot grouping methods outlined in Li et al. (2024) to compare results on Mixtral 8x7B, and report the results in Table 6. Among the similarity metrics evaluated, *router-logits* exhibits the poorest performance, indicating its unsuitability for task-agnostic settings due to its reliance on dataset-specific statistics. In both the 25% and 50% parameter reduction scenarios, all one-shot grouping methods underperform compared to *O-prune* presented in Table 3. This observation suggests that these grouping methods fail to form effective clusters, and can potentially result in lower performance

Table 6: Comparisons for different similarity metric to single-shot grouping method and our HC-SMoE on Mixtral 8x7B with reducing experts to average 6 and 4 per layer.

| Model | Metric | ARC-c | ARC-e | BoolQ | HellaSwag | MMLU | OBQA | RTE | Winogrande | Average |
|---|---|---|---|---|---|---|---|---|---|---|
| Mixtral 8x7B | None | 0.5648 | 0.8422 | 0.8505 | 0.6490 | 0.6712 | 0.350 | 0.7112 | 0.7593 | 0.6748 |
| Mixtral 6x7B | *router-logits* | 0.2619 | 0.5564 | 0.5208 | 0.432 | 0.2503 | 0.194 | 0.5271 | 0.5848 | 0.4159 |
| | *weight* | 0.4974 | 0.7955 | 0.781 | 0.6131 | 0.5244 | 0.34 | **0.6715** | **0.7585** | 0.6227 |
| | *expert-output* | 0.506 | **0.8056** | 0.8373 | 0.613 | 0.5595 | 0.306 | 0.6318 | 0.7474 | 0.6258 |
| | HC-SMoE | **0.5145** | 0.8043 | **0.8554** | **0.6142** | **0.6043** | **0.324** | **0.6715** | 0.7514 | **0.6425** |
| Mixtral 4x7B | *router-logits* | 0.2116 | 0.2765 | 0.4954 | 0.2767 | 0.2452 | 0.108 | 0.4910 | 0.4964 | 0.3251 |
| | *weight* | 0.4172 | 0.7382 | 0.7862 | 0.5457 | 0.4223 | 0.256 | 0.5523 | 0.7143 | 0.554 |
| | *expert-output* | 0.4326 | 0.7386 | **0.8021** | 0.5467 | 0.429 | **0.278** | **0.5704** | 0.7245 | 0.5652 |
| | HC-SMoE | **0.4573** | **0.7454** | 0.8018 | **0.5709** | **0.4571** | 0.270 | 0.5523 | **0.7285** | **0.5729** |

Table 7: Various merging methods with hierarchical clustering average linkage based on expert outputs. **Fix-Dom** represents fixed-dominant merging described in Section 3.2.3. **Avg** in the Merge column denotes the average score among all the merging strategy under same model settings.

| Model | Merge | ARC-c | ARC-e | BoolQ | HellaSwag | MMLU | OBQA | RTE | Winogrande | Average |
|---|---|---|---|---|---|---|---|---|---|---|
| Qwen 60x2.7B | None | 0.3951 | 0.7012 | 0.8135 | 0.5932 | 0.6047 | 0.310 | 0.7329 | 0.6559 | 0.6008 |
| Qwen 45x2.7B | Frequency | 0.3660 | 0.6578 | 0.7948 | 0.5520 | 0.5332 | **0.272** | **0.7509** | 0.6464 | **0.5716** |
| | Average | 0.3584 | 0.6553 | **0.7936** | 0.5516 | **0.5348** | 0.270 | 0.7473 | **0.6559** | 0.5709 |
| | Fix-Dom | **0.3695** | **0.6692** | 0.7896 | **0.5555** | 0.5338 | 0.262 | 0.7365 | 0.6535 | 0.5712 |
| | Avg | 0.3646 | 0.6608 | 0.7927 | 0.5530 | 0.5339 | 0.268 | 0.7449 | 0.6519 | 0.5712 |
| Qwen 30x2.7B | Frequency | **0.3532** | **0.6149** | 0.7535 | **0.4695** | **0.4534** | **0.228** | 0.6606 | 0.6456 | 0.5223 |
| | Average | **0.3575** | 0.6145 | **0.7554** | 0.4706 | 0.4531 | **0.228** | **0.6643** | 0.6488 | **0.5240** |
| | Fix-Dom | 0.3439 | 0.6132 | 0.7544 | 0.4679 | 0.4445 | **0.228** | **0.6643** | **0.6504** | 0.5208 |
| | Avg | 0.3515 | 0.6142 | 0.7544 | 0.4693 | 0.4503 | **0.228** | 0.6631 | 0.6483 | 0.5224 |

even when attempting to absorb all expert knowledge. The method based on the expert output metric demonstrates superior performance over other similarity metrics. It outperforms router-logits by 24.01% and weights by 1.12% when reducing 50% of the expert parameters. This finding highlights the importance of selecting appropriate similarity metrics for effective expert grouping. HC-SMoE demonstrates a clear advantage over the one-shot grouping approaches. It achieves average improvements of 1.98% and 1.67% in the 25% and 50% parameter reduction settings, respectively.

**Ablation on Different Merging Methods.** Table 7 presents the results of hierarchical clustering with three merging strategies: *frequency*, *average*, and *fixed-dominant merging*. For Qwen30x2.7B, the average merging method demonstrates superior performance. It exceeds frequency merging by 0.17% and marginally enhances overall performance. This outcome substantiates our assertion that once a high-quality cluster is identified, the specific merging method becomes modestly influential on the final performance. The rationale behind this phenomenon lies in the functional similarity exhibited by experts within the same group, as evidenced by their similar outputs. Thus, the model maintains robust performance irrespective of the merging strategy employed. It is noteworthy that all three merging methods outperform the four baselines in Table 2. This observation further substantiates the effectiveness of HC-SMoE in preserving model performance during the merging process.

## 5   CONCLUSION

In this paper, we presented HC-SMoE, a retraining-free, task-agnostic, and scalable expert merging framework that employed hierarchical clustering to reduce the parameters of SMoE models. By employing on expert outputs as the similarity metric and leveraging hierarchical clustering, HC-SMoE effectively captured functional similarities between experts, surpassing previous merging and pruning methods. Our comprehensive evaluation on two representative large-scale models, Qwen and Mixtral, demonstrated that HC-SMoE retained the models' general language abilities even when significantly reducing the number of experts. The experimental results also validated the robustness and scalability of our approach. HC-SMoE achieved notable improvements over existing baselines. This work not only provided a practical solution for optimizing SMoE models but also opened up a broader domain for further research on task-agnostic model compression strategies for SMoE.

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

# A EXPLORATORY EXPERIMENTS

All the experiments code is released at `https://anonymous.4open.science/r/TAMP-11E2`.

## A.1 NON-UNIFORM HIERARCHICAL CLUSTERING

In our main experiments, the number of clusters in each layer is fixed and uniform due to model design choices. Here, we explore a more flexible approach that allows different numbers of clusters in each layer while maintaining an overall 25% or 50% reduction of experts. To determine the cluster count per layer, we first select the top $r\%$ most frequently activated experts based on their activation frequencies across layers. We then count the number of these experts that remain in each layer to guide the selection of clusters in that layer, followed by hierarchical clustering.

For example, in the uniform clustering setting for Qwen with a 25% reduction, the distribution will be $[45, 45, 45, 45, ..., 45]$ across all layers. In contrast, the non-uniform setting might result in a distribution like $[48, 45, 40, 42, 50, ...]$, as long as the overall number of clusters aligns with the target reduction. Table 8 presents the results of this non-uniform clustering strategy.

## A.2 FIXED-DOMINANT MERGING

The fixed-dominant (Fix-Dom) merging approach modifies the traditional ZipIt (Stoica et al., 2024) feature similarity calculation. Rather than concatenating features from all experts and computing pairwise correlations, we fix the feature order of a designated dominant expert as a reference point. Correlations are then computed between this fixed order and the features of other experts, as shown in Figure 3. Features from secondary experts are grouped with their most correlated counterparts in the dominant expert. The merging process then applies an appropriate weighting scheme, such as average merging, preserving the dominant expert's weight feature order while simplifying the merging process.

Feature similarity is defined as the pairwise correlation between these output features, using formulas adapted from (Li et al., 2016). In the original ZipIt model merging, output features are taken after each linear layer. However, since we aim to merge entire experts, each containing three linear layers, we use the intermediate activation features, which is the activations after the non-linear function and before feeding into $W_{down}$: act $= (xW_{gate}) \odot xW_{up}$ to compute similarity. This approach considers expert similarity from an activation perspective, but we can also use the experts' weights as the "feature" for correlation or even combine both activation and weight features.

The Fix-Dom merging technique has two main advantages: it preserves the structural integrity of the dominant expert's feature arrangement and accelerates the merging process compared to the original ZipIt method. Instead of iteratively selecting and merging highly correlated features until the target dimension is reached, fix-dom merge performs a more efficient grouping. For example, in Mixtral8x4B, ZipIt takes approximately 725 minutes, while Fix-Dom merge completes in just 7 minutes, making it over 100 times faster. For performance comparisons between ZipIt and fix-dom merge using various feature selections (activation, weight, and activation + weight), refer to Table 9.

Table 8: The comparison between ZipIt and Fix-Dom merging for reducing 25% experts of Qwen under the same expert clustering groups.

| Linkage | Metric | Merge | ARC-c | ARC-e | BoolQ | HellaSwag | MMLU | OBQA | RTE | Winogrande | Average |
|---|---|---|---|---|---|---|---|---|---|---|---|
| Single | *weight* | Freq | 0.2108 | 0.3493 | 0.5086 | 0.4536 | 0.2296 | 0.170 | 0.5596 | 0.5801 | 0.3827 |
| | | Fix-Dom | 0.2133 | 0.3531 | 0.4847 | 0.4588 | 0.2303 | 0.168 | 0.6101 | 0.5714 | 0.3862 |
| | *expert-output* | Freq | 0.3686 | 0.6604 | 0.7960 | 0.5587 | 0.5290 | 0.254 | 0.7401 | 0.6543 | 0.5701 |
| | | Fix-Dom | 0.3660 | 0.6612 | 0.7917 | 0.5564 | 0.5302 | 0.262 | 0.7292 | 0.6527 | 0.5687 |
| Average | *weight* | Freq | 0.2125 | 0.3535 | 0.5024 | 0.4543 | 0.2287 | 0.174 | 0.5560 | 0.5785 | 0.3825 |
| | | Fix-Dom | 0.2116 | 0.3497 | 0.4951 | 0.4565 | 0.2327 | 0.164 | 0.5921 | 0.5738 | 0.3844 |
| | *expert-output* | Freq | 0.3575 | 0.6561 | 0.7933 | 0.5538 | 0.5319 | 0.272 | 0.7365 | 0.6551 | 0.5695 |
| | | Fix-Dom | 0.3558 | 0.6582 | 0.7917 | 0.5558 | 0.5306 | 0.270 | 0.7256 | 0.6559 | 0.5680 |

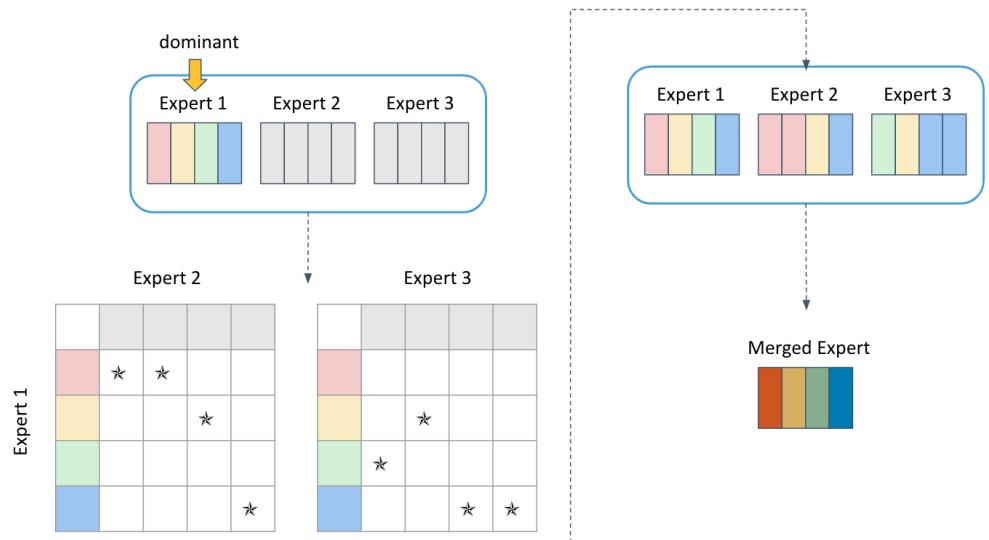

Figure 3: Fixed-Dominant Merging. Given a group of experts, the dominant expert's feature order is fixed. The feature correlations with other non-dominant experts are calculated to align their features with those of the dominant expert. The merging process then combines these aligned features to form the final merged expert.

Table 9: The comparison between ZipIt and Fix-Dom merging for reducing Mixtral 8x7B to Mixtral 4x7B under the same expert clustering groups.

| Feature | Merge | ARC-c | ARC-e | BoolQ | HellaSwag | MMLU | OBQA | RTE | Winogrande | Average |
|---|---|---|---|---|---|---|---|---|---|---|
| act | zipit | 0.3959 | **0.6978** | 0.7352 | 0.5350 | 0.4256 | 0.252 | 0.5776 | 0.7080 | 0.5409 |
| | Fix-Dom | **0.4036** | 0.6873 | **0.7951** | **0.5351** | **0.4471** | **0.278** | **0.6462** | **0.7174** | **0.5637** |
| weight | zipit | 0.3959 | 0.7062 | 0.7976 | 0.5376 | 0.4318 | 0.266 | **0.5848** | 0.6993 | 0.5524 |
| | Fix-Dom | **0.4334** | **0.7290** | **0.8009** | **0.5608** | **0.4913** | **0.280** | 0.5596 | **0.7253** | **0.5725** |
| act+weight | zipit | 0.4078 | 0.7146 | **0.8125** | 0.5389 | 0.4364 | **0.270** | 0.5921 | 0.7009 | 0.5592 |
| | Fix-Dom | **0.4283** | **0.7184** | 0.7774 | **0.5501** | **0.4737** | 0.264 | 0.5921 | **0.7388** | **0.5679** |

## B    EFFICIENCY DISCUSSION

We evaluate computational and memory costs on the Mixtral 8x7B and Qwen1.5-MoE-A2.7B-Chat models in both their original and merged versions. All experiments use the same calibration dataset as the main experiments and consist of 32 sequences of 2048 tokens sampled from the C4 corpus (Raffel et al., 2020). The results in Table 10 show that a reduction in the number of experts leads to significant decreases in memory usage and GLOPs without impact on throughput and latency. The ideal benefits of reduced router latency from fewer output channels are not realized since we retain the original router weights to prevent accuracy degradation. As a result, the router functions as if the original number of experts exists, with experts within the same group producing identical outputs through their corresponding merged experts.

## C    FREQUENCY ANALYSIS

### C.1    MIXTRAL 8X7B

We present the activation frequency analysis of all experts in Mixtral 8x7B (Jiang et al., 2024) using our sampling dataset from C4 (Raffel et al., 2020) and eight language benchmarks. The results provide evidence against using frequency as the sole criterion for determining the number of

Table 10: Evaluation of computational and memory efficiency across multiple models. For Mixtral: Mixtral 8x7B (original), Mixtral 6x7B (25% pruned), and Mixtral 4x7B (50% pruned). For Qwen1.5-MoE-A2.7B-Chat: Qwen 60x2.7B (original), Qwen 45x2.7B (25% pruned), and Qwen 30x2.7B (50% pruned). All measurements use identical input sequences and include throughput (tokens per ms), latency (s), GFLOPs, model memory, and model size (number of parameters).

| Models | Throughput | Latency | GFLOPs | Memory | Model Size |
|--------|-----------|---------|--------|--------|-----------|
| Mixtral 8x7B | $13.45 \pm 1.30$ | $2.854 \pm 0.333$ | 2989 | 87.49GB | 46.7B |
| Mixtral 6x7B | $13.87 \pm 0.47$ | $2.666 \pm 0.093$ | 2267 | 66.49GB | 35.4B |
| Mixtral 4x7B | $13.96 \pm 0.65$ | $2.599 \pm 0.166$ | 1546 | 45.49GB | 24.2B |
| Qwen 60x2.7B | $24.08 \pm 0.17$ | $1.593 \pm 0.168$ | 916 | 27.04GB | 14.3B |
| Qwen 45x2.7B | $23.95 \pm 0.24$ | $1.541 \pm 0.011$ | 717 | 21.23GB | 11.2B |
| Qwen 30x2.7B | $23.16 \pm 0.42$ | $1.583 \pm 0.034$ | 518 | 15.44GB | 8.1B |

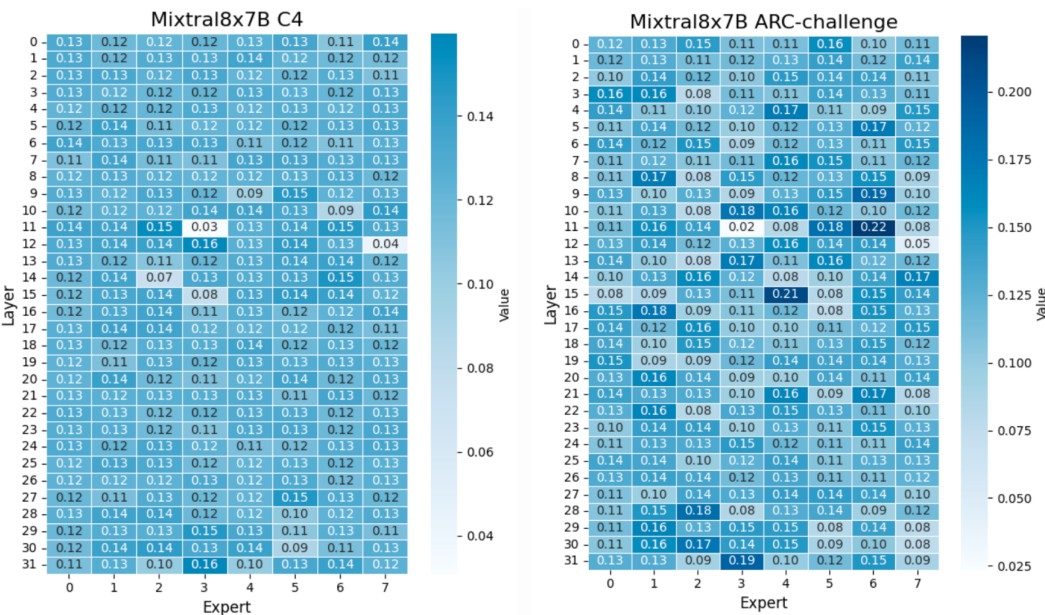

Figure 4: The frequency anslysis of Mixtral 8x7B on ARC-c and our sampling dataset of C4.

experts in each layer. The analysis reveals variability in activation frequency across different tasks, highlighting the fact that this metric is not a consistent or reliable indicator for expert selection in task-agnostic settings.

## C.2 TINYLLAMA-4X1.1B-MOE

The activation frequency analysis of all experts in TinyLLaMa-4x1.1B-MoE [1] on our sampling dataset of C4 (Raffel et al., 2020) and eight language benchmarks. It can be the evidence of poor expert utilization in SMoE, since one of the experts is seldom chosen among all tasks.

---

[1] https://huggingface.co/s3nh/TinyLLama-4x1.1B-MoE

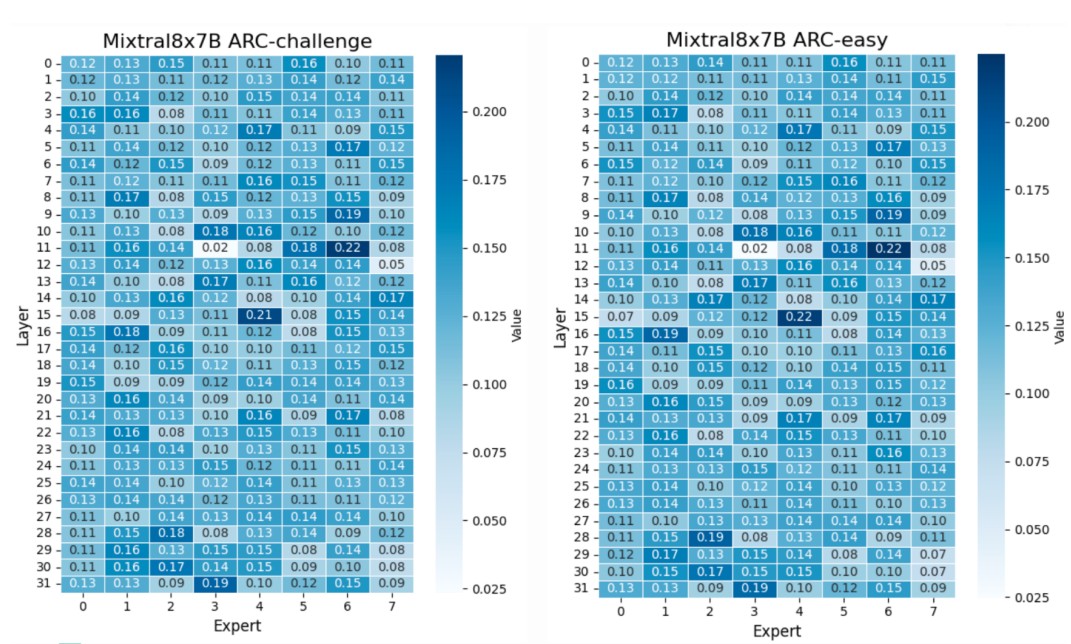

Figure 5: The frequency anslysis of Mixtral 8x7B on ARC-c and ARC-e.

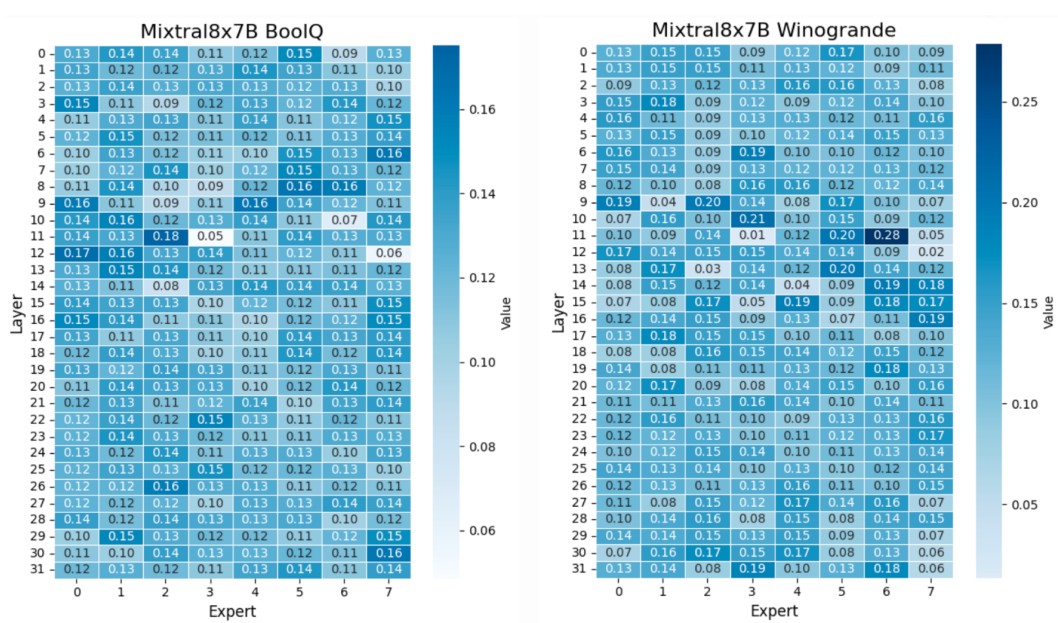

Figure 6: The frequency anslysis of Mixtral 8x7B on BoolQ and Winogrande.

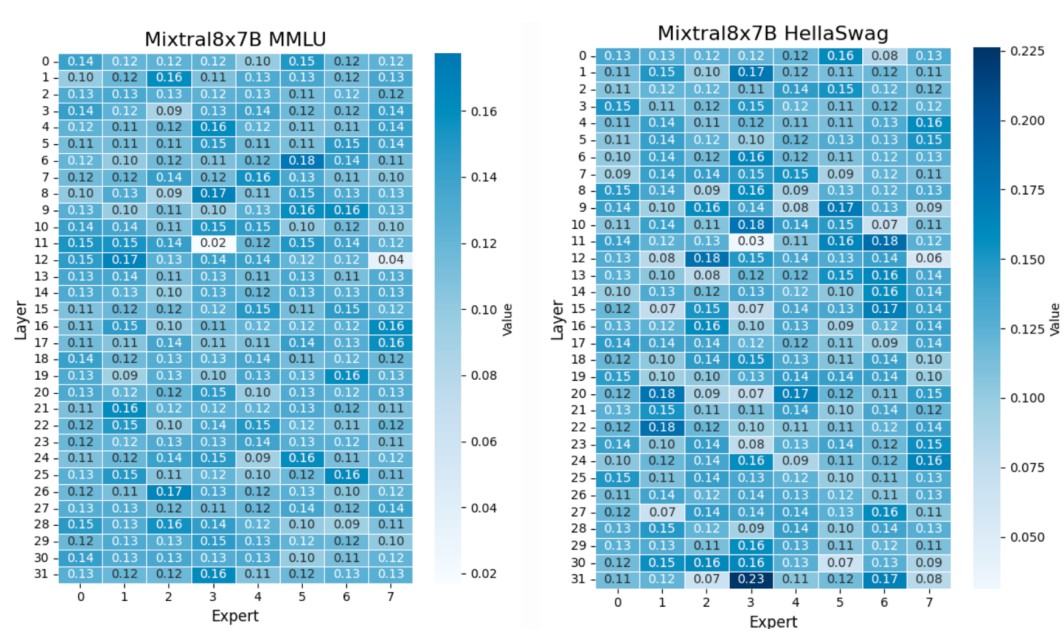

Figure 7: The frequency anslysis of Mixtral 8x7B on MMLU and HellaSwag.

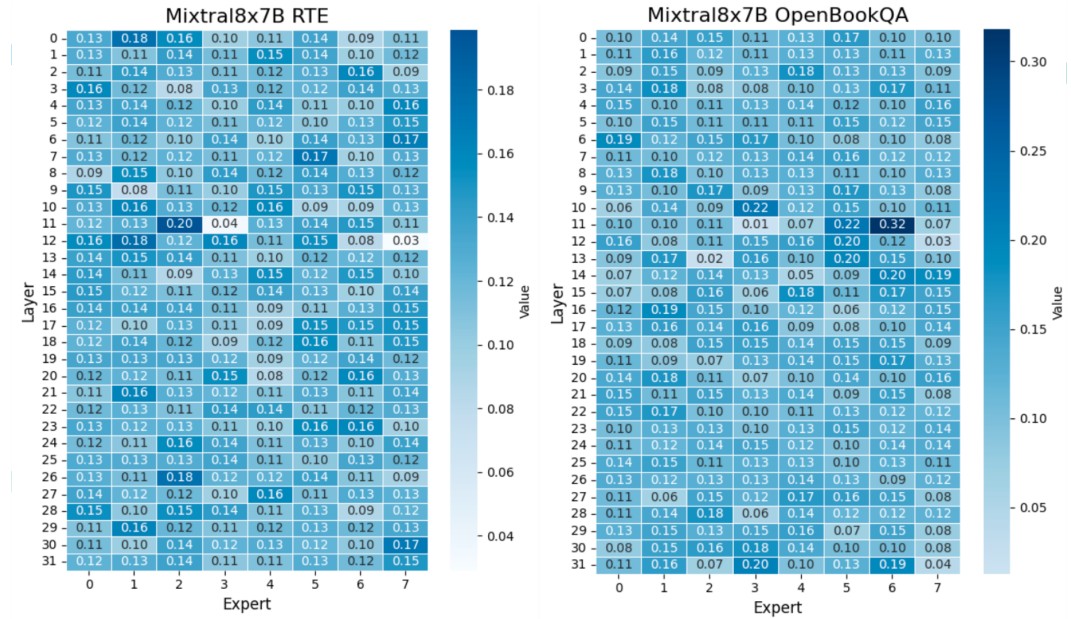

Figure 8: The frequency anslysis of Mixtral 8x7B on RTE and OpenBookQA.

