# OpenReview forum: "Retraining-Free Merging of Sparse Mixture-of-Experts via Hierarchical Clustering"
_ICLR.cc/2025/Conference — ICLR 2025 Conference Withdrawn Submission_

### Official Review · Reviewer_G91C · 2024-10-18

**Soundness:** 3
**Presentation:** 3
**Contribution:** 3
**Rating:** 6
**Confidence:** 4

**Summary:**

This paper introduces Hierarchical Clustering for Sparsely Activated Mixture of Experts (HC-SMoE), a task-agnostic framework for merging experts within an SMoE model. HC-SMoE aims to reduce the model's parameters without requiring retraining. Experiment results on a series of benchmarks show its effectiveness.

**Strengths:**

1) Pruning experts in MoE models can indeed reduce the difficulty of deployment.
2) The paper is easy to follow, and the ablation study is comprehensive.
3) The experimental results on Qwen and Mixtral are convincing.

**Weaknesses:**

1) O-prune [1] requires enumerating all possible combinations of experts, resulting in significant time overhead. I would like to know how HC-SMoE compares to other approaches in terms of runtime and resource consumption.
2) O-prune [1] also conducts experiments on domain-specific tasks (e.g., GSM8K, Math). I am interested in the performance of HC-SMoE on these datasets.

[1] Lu, Xudong et al. “Not All Experts are Equal: Efficient Expert Pruning and Skipping for Mixture-of-Experts Large Language Models.” Annual Meeting of the Association for Computational Linguistics (2024).

**Questions:**

See Weaknesses above.

---

> ### Author Response · Authors · 2024-12-04
> **Rebuttal by Authors - Response to Comment 1**
>
> **Rebuttal:**
>
> We appreciate the reviewer’s valuable feedback and effort spent on the review and would like to respond to the reviewer’s questions as follows.
>
> **C1.** O-prune [1] requires enumerating all possible combinations of experts, resulting in significant time overhead. I would like to know how HC-SMoE compares to other approaches in terms of runtime and resource consumption.
>
> **Response:** We appreciate the reviewer's interest in the runtime and resource consumption comparison of HC-SMoE with other approaches. The following sections provide a detailed analysis of the computational requirements.
>
> The experimental evaluation of HC-SMoE utilized five L48 GPUs for Mixtral and four V100 GPUs for Qwen. The reported runtimes exclude model loading and dataset preparation overhead to focus on algorithmic efficiency. The results are presented in Tables r9 and r10.
>
> O-prune requires exhaustive evaluation of expert combinations, which induces substantial computational overhead. This limitation becomes more pronounced as the number of experts increases in MoE architectures. The computational constraints restricted our evaluation of O-prune on Qwen to 100 random combinations per layer for performance estimation.
>
> In contrast, HC-SMoE completes its operations in under ten minutes for comparable settings. HC-SMoE exhibits slightly longer runtimes than F-prune, S-prune, and M-SMoE due to the hierarchical clustering process, which requires iterative distance recalculations and cluster groupings. Nevertheless, HC-SMoE demonstrates substantial efficiency advantages over O-prune. For example, O-prune requires approximately one hour to achieve a 50% pruning ratio on Mixtral, whereas HC-SMoE completes the same task in under two minutes.
>
> | Model         | Method   | Runtime (s) | Memory (GB) |
> |---------------|----------|-------------|-------------|
> | Mixtral 6x7B  | F-prune  | 65.482      | 106.601    |
> |               | S-prune  | 63.176      | 106.593    |
> |               | O-prune  | 1890.651    | 122.640     |
> |               | M-SMoE   | 46.933      | 106.601    |
> |               | HC-SMoE  | 110.899     | 138.039    |
> | Mixtral 4x7B  | F-prune  | 63.004      | 106.601    |
> |               | S-prune  | 63.818      | 106.593    |
> |               | O-prune  | 3605.170    | 122.640     |
> |               | M-SMoE   | 49.835      | 106.601    |
> |               | HC-SMoE  | 110.745     | 138.039    |
>
> **Table r9.** Computational costs of pruning Mixtral 8x7B to average six (25% pruning ratio) and four (50% pruning ratio) experts per layer for HC-SMoE and four baselines.
>
> | Model          | Method        | Runtime (s) | Memory (GB) |
> |----------------|---------------|-------------|-------------|
> | Qwen 45x2.7B   | F-prune       | 95.04       | 61.605     |
> |                | S-prune       | 94.56       | 61.605     |
> |                | O-prune (100) | 824.47      | 70.849      |
> |                | M-SMoE        | 107.46      | 48.829      |
> |                | HC-SMoE       | 290.38      | 48.701     |
> | Qwen 30x2.7B   | F-prune       | 94.84       | 61.605     |
> |                | S-prune       | 94.89       | 61.605     |
> |                | O-prune (100) | 840.24      | 70.849      |
> |                | M-SMoE        | 107.20      | 48.829      |
> |                | HC-SMoE       | 323.04      | 48.701     |
>
> **Table r10.** Computational costs of pruning Qwen1.5-MoE-A2.7B-Chat to average 45 (25% pruning ratio) and 30 (50% pruning ratio) experts per layer for HC-SMoE and four baselines.

---

> ### Author Response · Authors · 2024-12-04
> **Rebuttal by Authors - Response to Comment 2**
>
> **C2.** O-prune [1] also conducts experiments on domain-specific tasks (e.g., GSM8K, Math). I am interested in the performance of HC-SMoE on these datasets.
>
> **Response:** We appreciate the reviewer's interest in HC-SMoE's performance on domain-specific datasets.
>
> Our experimental results indicate that HC-SMoE achieves comparable performance to O-prune on GSM8K dataset, which substantiates the effectiveness of our methodology. HC-SMoE accomplishes these results with significant computational efficiency, and requires less than 1/10 of the runtime compared to O-prune when performing on Mixtral models. Furthermore, when applied to models with increased expert counts, such as Qwen, HC-SMoE maintains optimal performance within practical timeframes, which demonstrates its scalability and effectiveness for complex MoE architectures. The detailed experimental results are presented in Table r11 for Qwen models and Table r12 for Mixtral models.
>
> | Model         | Method        | Calib Dataset | GSM8K (5-shot) - strict | GSM8K (5-shot) - flexible | Runtime (s) |
> |---------------|---------------|---------------|-------------------------|---------------------------|-------------|
> | Qwen 60x2.7B  |               |               | 0.3685                  | 0.5155                    |             |
> | Qwen 45x2.7B  | O-prune (100) | C4            | 0.2904                  | 0.3101                    | 129.41     |
> |               | HC-SMoE       | C4            | 0.2835                  | 0.3139                    | 290.38     |
> |               | O-prune (100) | MATH          | 0.2721                  | 0.3313                    | 161.05     |
> |               | HC-SMoE       | MATH          | **0.4124**                  | **0.4837**                    | 305.24     |
> | Qwen 30x2.7B  | O-prune (100) | C4            | 0.0743                  | 0.0766                    | 130.40     |
> |               | HC-SMoE       | C4            | 0.0766                  | 0.0788                    | 323.04     |
> |               | O-prune (100) | MATH          | 0.0295                  | 0.0409                    | 154.88     |
> |               | HC-SMoE       | MATH          | **0.3594**                  | **0.3768**                    | 293.12     |
>
> **Table r11.** Comparison of 5-shot GSM8k accuracy between HC-SMoE and O-Prune-100 on Qwen1.5-MoE-A2.7B-Chat  using strict and flexible matching across different calibration datasets. Please note that due to Qwen's large expert count, evaluation of O-prune was limited to 100 expert combinations per layer, as exhaustive computation would require prohibitive runtime for practical deployment.
>
> | Model        | Method      | Calib_Dataset | gsm8k (5-shot) - strict | gsm8k (5-shot) - flexible | Runtime    |
> |--------------|-------------|---------------|-------------------------|---------------------------|------------|
> | Mixtral 8x7B |             |               | 0.5876                  | 0.5898                    |            |
> | Mixtral 6x7B | O-prune     | C4            | 0.4405                  | 0.4435                    | 1890.651  |
> |              | HC-SMoE     | C4            | 0.3874                  | 0.3889                    | **110.899**   |
> |              | O-prune     | MATH          | **0.4996**                  | **0.5034**                   | 1901.067  |
> |              | HC-SMoE     | MATH          | 0.4678                  | 0.4716                    | **111.732**   |
> | Mixtral 4x7B | O-prune     | C4            | 0.2381                  | 0.2411                    | 3605.170  |
> |              | HC-SMoE     | C4            | 0.1812                  | 0.1827                    | **110.745**   |
> |              | O-prune     | MATH          | **0.3715**                  | **0.3738**                    | 3628.349  |
> |              | HC-SMoE     | MATH          | 0.2654                  | 0.2691                    | **103.249**   |
>
> **Table r12.** Comparison of 5-shot GSM8k accuracy between HC-SMoE and O-Prune on Mixtral 8x7B using strict and flexible matching across different calibration datasets. Please note that the computational requirements of O-prune exceed other methods by a factor of ten to thirty, which presents significant practical limitations for model compression tasks.

---

### Official Review · Reviewer_A87Z · 2024-10-28

**Soundness:** 3
**Presentation:** 3
**Contribution:** 2
**Rating:** 5
**Confidence:** 3

**Summary:**

This paper proposed a new expert merging framework, named Hierarchical Clustering for Sparsely activated Mixture of Experts (HC-SMoE), to reduce SMoE model parameters without retraining. The proposed method is simple but effective, and the experiments demonstrated the efficacy of the proposed method.

**Strengths:**

* The proposed method is simple but effective,
* The paper is very easy to follow.
* The experiments are comprehensive and the results are very promising.

**Weaknesses:**

* The motivation of using the "Hierarchical" clustering is not clear to me. I cannot intuitively get the idea of why hierarchical clustering is better than simple K-means clustering, although the results confirmed that K-means clustering is less effective. Besides, the paper proposed to use a "hard" hierarchical clustering, and I am wondering if it is more effective to use "soft" hierarchical clustering or simply "soft" clustering without hierarchies.
* The choice of the calibration dataset. I did not see any ablation study about the choice of the calibration dataset, and I think the performance of the proposed method should highly depend on the calibration dataset. If the calibration dataset is not comprehensive enough, e.g., not covering enough domain specific data, the clustering may not be very informative, which may lead to poor performance. For example, if we want the LLM to perform well on a law-related or medical-related tasks, can you also rely on the same calibration dataset used in the experiments?
* Some minor issues:
  * In Fig. 1, why you did not compare the methods on 14B model?
  * Section 3.2.1 presents the method of similarity metric but contains a lot of discussions about related work.
  * In line 299/300, is alpha_i fixed or not? If it is fixed, will it also suffer from the issue that you mentioned in line 199-203 about frequency-based method?
  * In Table 4, the best performance of 'ARC-c' should be the Average linkage using the Weight setting, right?

**Questions:**

Please refer to my question above.

---

> ### Author Response · Authors · 2024-12-04
> **Rebuttal by Authors - Response to Comment 1**
>
> **Rebuttal:** We appreciate the reviewer’s valuable feedback and effort spent on the review and would like to respond to the reviewer’s questions as follows.
>
> **C1.** The motivation of using the "Hierarchical" clustering is not clear to me. I cannot intuitively get the idea of why hierarchical clustering is better than simple K-means clustering, although the results confirmed that K-means clustering is less effective.
>
> **Response:** We appreciate the reviewer for raising the question about the motivation behind using hierarchical clustering instead of K-means. We elaborate on this by providing a detailed explanation structured into two parts: **theoretical justification** and **experimental validation**.
>
> ***Theoretical Justification***
>
> We selected hierarchical clustering over K-means based on two primary reasons: its stability and determinism. The initialization of cluster centroids in K-means is often random, which can result in different clustering outcomes across multiple runs on the same dataset [1]. This non-deterministic behavior of K-means makes it less suitable for tasks where the reproducibility and consistency of clustering results are crucial, such as in the evaluation of downstream tasks. In contrast, hierarchical clustering generates deterministic results for a given dataset and linkage method. This deterministic property ensures that the clustering results are reproducible and consistent across different runs. Furthermore, hierarchical clustering employs a systematic approach to merge clusters based on a specified linkage criterion. This linkage criterion determines the distance between clusters and governs the merging process. By optimizing the linkage criterion, hierarchical clustering guarantees the formation of stable clusters that minimize the intra-cluster distance and maximize the inter-cluster distance. This optimization ensures that the resulting clusters are compact and well-separated, which is desirable for effective pruning and merging of experts in the context of model compression.
>
> ***Experimental Validation***
>
> In addition to the theoretical justification, we further validated the effectiveness of hierarchical clustering compared to K-means through experiments. We conducted experiments using both methods across three similarity metrics (i.e., router logits, expert outputs, and expert weights) and evaluated six clustering criteria:
> 1. **L2 distance**: $||T(x)−S(x)||_2$, where $T(x)$ and $S(x)$ represent the outputs of the original and pruned models, respectively. Lower values are better.
> 2. **Cosine similarity**: $\text{cosine-similarity}(T(x),S(x))$. Higher values are better.
> 3. **Silhouette score (Euclidean)**: Measures how similar an object is to its cluster compared to other clusters, using Euclidean distance. Higher values are better.
> 4. **Dunn index (Euclidean)**: Evaluates cluster compactness and separation, using Euclidean distance. Higher values are better.
> 5. **Silhouette score (Cosine)**: Similar to (3) but based on cosine similarity. Higher values are better.
> 6. **Dunn index (Cosine)**: Similar to (4) but based on cosine similarity. Higher values are better.
>
> Silhouette score evaluates clustering quality at the data point level ( i.e., expert level), while the Dunn index evaluates it at the cluster level. The Dunn index considers maximum intra-cluster and minimum inter-cluster distances, while the Silhouette score uses mean distances. Both metrics highlight clustering compactness and separability. We excluded evaluations involving the cosine similarity of expert weights due to the high computational cost of processing concatenated weight tensors, which would require excessive GPU memory. The detailed formulation is provided at the bottom of this response.
>
> Table r5 summarizes the results. Hierarchical clustering with expert outputs achieves the lowest L2 error and the highest cosine similarity with the original model outputs at 25% and 50% pruning ratios. Moreover, hierarchical clustering consistently outperforms K-means across most clustering metrics, demonstrating better clustering quality. These results substantiate the stability and effectiveness of hierarchical clustering in producing compact, well-separated clusters. Furthermore, the zero-shot performance on eight language tasks, as demonstrated in Table 5 in our manuscript, further supports the superiority of hierarchical clustering. Across all tasks, hierarchical clustering consistently outperforms K-means, and achieves better and more stable accuracy.
>
> Based on the above reasons, hierarchical clustering is preferable for its deterministic nature, superior clustering quality, and consistent performance across similarity metrics and benchmarks. We hope this explanation clarifies the advantages of hierarchical clustering over K-means.

---

> > ### Author Response · Authors · 2024-12-04
> > **Rebuttal by Authors - Response to Comment 1 (cont.)**
> >
> > | Model            | Cluster      | Metric        | L2 error  | Cosine Similarity | Silhouette-Euc | Dunn-Euc | Silhouette-Cos | Dunn-Cos |
> > |------------------|--------------|-----------------|-----------|-------------------|----------------|----------|----------------|----------|
> > | Qwen 45x2.7B     | Hierarchical | *expert-output* | **3,806.8332** | **0.9972**           | **0.7909**         | **0.8252**   | **0.709**          | 0.5136   |
> > |                  | Kmeans       | *expert-output* | 6,769.3674| 0.991             | 0.6093         | 0.2145   | 0.6489         | **0.53**     |
> > |                  | Hierarchical | *weight*        | 9,307.8344| 0.9874            | **0.7358**         | **0.9801**   | None           | None     |
> > |                  | Kmeans       | *weight*        | 7,962.1627| 0.9919            | 0.6125         | 0.9225   | None           | None     |
> > |                  | Hierarchical | *router-logits* | 7,572.2246| 0.9897            | **0.6688**         | **0.8453**   | **0.7169**         | **0.5204**   |
> > |                  | Kmeans       | *router-logits* | 7,463.5265| 0.99              | 0.6196         | 0.481    | 0.6389         | 0.3469   |
> > | Qwen 30x2.7B     | Hierarchical | *expert-output* | **8,142.6961** | **0.9878**            | **0.6104**         | **0.8126**   | **0.4233**         | 0.4939   |
> > |                  | Kmeans       | *expert-output* | 9,796.4269| 0.9842            | 0.1851         | 0.221    | 0.2375         | **0.5136**   |
> > |                  | Hierarchical | *weight*        | 11,400.2119| 0.9791           | **0.4876**         | **0.9734**   | None           | None     |
> > |                  | Kmeans       | *weight*        | 10,894.4059| 0.9808           | 0.238          | 0.9245   | None           | None     |
> > |                  | Hierarchical | *router-logits* | 10,022.4158| 0.9826           | **0.4495**         | **0.5536**   | **0.4673**         | **0.2115**   |
> > |                  | Kmeans       | *router-logits* | 10,348.0981| 0.9825           | 0.2566         | 0.3731   | 0.2653         | 0.203    |
> >
> > **Table r5.** The ablation study of measuring the error of the last layer with the original model, and the cluster quality with different clustering methods and similarity metrics on the Qwen model. We conduct the measurement on Qwen 45x2.7B (pruning ratio 25%) and Qwen 30x2.7B (pruning ratio 50%). We bold the score of highest Silhouette score and Dunn Index every two row because these two criteria cannot directly compared to methods using different metrics, i.e., methods using expert outputs cannot compare with methods using weights.

---

> ### Author Response · Authors · 2024-12-04
> **Rebuttal by Authors - Response to Comment 1 (cont.)**
>
> The following equations formulate the Silhouette score and Dunn index:
>
> **Silhouette Score**
>
> For each data point $i$ in cluster $C_I$, let $a(i)$ denote the mean distance between $i$ and all other data points in the same cluster, and let $b(i)$ represent the smallest mean distance of $i$ to all points in any other cluster, defined as:
>
> \begin{align}
> a(i) = \frac{1}{|C_I|-1}\sum_{j\in C_I, i\neq j}d(i, j), \quad
> b(i) = \min_{J\neq I}\frac{1}{|C_J|}\sum_{j\in C_J}d(i,j),
> \end{align}
>
> where $d(i, j)$ represents the distance function between data points $i$ and $j$. The Silhouette score of a single data point $i$ is formulated as follows:
>
> $$s(i)=\frac{b(i)-a(i)}{\max(a(i), b(i))},\quad \text{if }|C_i| > 1 , \quad s(i)=0, \quad \text{if }|C_I|=1$$
>
> To calculate a single Silhouette score for all experts, the mean of the Silhouette scores from all data points is used as the representative:
>
> \begin{equation}
> S = \frac{1}{n}\sum_{i=1}^n s(i).
> \end{equation}
>
> The range of Silhouette scores is from +1 to -1, with +1 indicating well-separated clusters.
>
> **Dunn Index**
>
> The Dunn Index is an internal evaluation scheme for measuring the quality of clustering algorithms. Let $C_I$ be a cluster of vectors and $x, y$ be any two feature vectors. The intra-cluster distance is defined as follows:
>
> \begin{equation}
> \Delta_I = \max_{x,y\in C_I}d(x, y)
> \end{equation}
>
> The inter-cluster distance is defined as follows:
>
> \begin{equation}
> \delta(C_I,C_J) = \min_{x\in C_I, y\in C_J}d(x, y)
> \end{equation}
>
> The Dunn Index equals the ratio of the minimum inter-cluster distance to the maximum intra-cluster distance for $m$ clusters:
>
> \begin{equation}
> DI_m = \frac{\min_{1\leq i < j \leq m}\delta(C_i, C_j)}{\max_{1\leq k\leq m}\Delta_k}
> \end{equation}
>
> These formulations provide a mathematical foundation for evaluating the quality of clustering results in the context of the ablation study.

---

> ### Author Response · Authors · 2024-12-04
> **Rebuttal by Authors - Response to Comment 2**
>
> **C2.** Besides, the paper proposed to use a "hard" hierarchical clustering, and I am wondering if it is more effective to use "soft" hierarchical clustering or simply "soft" clustering without hierarchies.
>
> **Response:** We appreciate the reviewer for raising the question of considering soft clustering in the MoE merging framework. To explore this, we implemented the Fuzzy C-Means (FCM) algorithm [1], which allows each expert to belong to multiple clusters with varying degrees of membership.
>
> Let $E = \{e_1, \ldots, e_n\}$ denote the $n$ experts to be clustered. The FCM algorithm outputs $c$ cluster centers $C = \{c_1, \ldots, c_c\}$ and a partition matrix. The degree of membership of expert $e_i$ in cluster $c_j$ is denoted by $u_{ij} \in [0, 1]$. FCM minimizes the following objective function:
> \\begin{equation}
> J_m = \sum_{i=1}^N \sum_{j=1}^C u_{ij}^m \|e_i - c_j\|^2 \quad \ldots \quad (1),
> \\end{equation}
> where the membership degrees and cluster centers are updated iteratively as follows:
> \\begin{align}
> u_{ij} = \left(\sum_{k=1}^C \left(\frac{\|e_i - c_j\|}{\|e_i - c_k\|}\right)^{\frac{2}{m-1}}\right)^{-1}, \quad
> c_j = \frac{\sum_{i=1}^N u_{ij}^m \cdot e_i}{\sum_{i=1}^N u_{ij}^m} \quad \ldots \quad (2)
> \\end{align}
> For our experiments, we set the hyperparameter $m = 2$.
>
> Nevertheless, applying soft clustering introduces ambiguity in our frequency-weighted merging method. To address this, we modified the merging process. For each cluster, the final merged expert is computed as a weighted sum of the experts, where the membership degree serves as the weight:
> \\begin{equation}
> e^{c_j} = \sum_{i=1}^n u_{ij} e_i \quad \ldots \quad (3)
> \\end{equation}
>
> Since HC-SMoE does not modify the router weights (as presented in Fig. 2(c) of the manuscript), input tokens are routed to the corresponding merged expert based on their original assignment. In the FCM setting, this direct routing becomes infeasible, as every expert belongs to all clusters to some degree. To adapt, we merged the router weights using the same weighted formula as above.
>
> Table r6 and r7 compare HC-SMoE with FCM. The results indicate a significant accuracy degradation when using FCM, which is likely due to interference in the router weights. This performance decline can be attributed to the fundamental differences between hard and soft clustering. In hard clustering, each expert belongs to exactly one cluster, which allows for a clear and unambiguous assignment of input tokens to merged experts. In contrast, soft clustering assigns each expert to multiple clusters with varying degrees of membership, which could lead to a more complex and potentially less effective routing process. Moreover, the weighted merging of router weights in the FCM setting may introduce noise and dilute the specialized knowledge captured by individual experts. This dilution can hinder the model's ability to effectively route input tokens to the most relevant experts, resulting in suboptimal performance.
>
> These findings highlight that applying soft clustering methods in the MoE merging framework requires a more sophisticated design to handle router weights, particularly in retraining-free settings. We consider it an interesting direction for future exploration to develop novel techniques that can leverage the benefits of soft clustering while mitigating the challenges associated with routing and merging in the context of MoE models.
>
> | Model           | Method        | ARC-c  | ARC-e  | BoolQ  | HellaSwag | MMLU   | OBQA   | RTE    | Winogrande | Average |
> |-----------------|---------------|--------|--------|--------|-----------|--------|--------|--------|------------|---------|
> | Qwen 60x2.7B    |               | 0.3951 | 0.7012 | 0.8135 | 0.5932    | 0.6047 | 0.31   | 0.7329 | 0.6559     | 0.6008  |
> | Qwen 45x2.7B | HC-SMoE   | **0.366**  | **0.6578** | **0.7948** | **0.552**     | **0.5332** | **0.272**  | **0.7509** | **0.6464**     | **0.5716**  |
> |    | Fuzzy-Cmeans  | 0.1954 | 0.282  | 0.4471 | 0.2707    | 0.2658 | 0.138  | 0.491  | 0.502      | 0.324   |
> | Qwen 30x2.7B | HC-SMoE   | **0.3532** | **0.6149** | **0.7535** | **0.4695**    | **0.4534** | **0.228**  | **0.6606** | **0.6456**     | **0.5223**  |
> |    | Fuzzy-Cmeans  | 0.1954 | 0.2816 | 0.4428 | 0.2708    | 0.2655 | 0.136  | 0.4946 | 0.5043     | 0.3239  |
>
> **Table r6.** Comparison of HC-SMoE and Fuzzy-Cmeans on Qwen1.5-MoE-A2.7B-Chat.
>
> [1]: Bezdek *et al*., FCM: The fuzzy c-means clustering algorithm, 1984

---

> > ### Author Response · Authors · 2024-12-04
> > **Rebuttal by Authors - Response to Comment 2 (cont.)**
> >
> > | Model           | Method        | Average | Winogrande | ARC-c  | ARC-e  | BoolQ  | HellaSwag | MMLU   | OBQA   | RTE    |
> > |-----------------|---------------|---------|------------|--------|--------|--------|-----------|--------|--------|--------|
> > | Mixtral 8x7B    |               | 0.6748  | 0.7593     | 0.5648 | 0.8422 | 0.8505 | 0.649     | 0.6712 | 0.35   | 0.7112 |
> > | Mixtral 6x7B | HC-SMoE   | **0.6425**  | **0.7514**     | **0.5145** | **0.8043** | **0.8554** | **0.6142** | **0.6043** | **0.324**  | **0.6715** |
> > |    | Fuzzy-Cmeans  | 0.6116  | 0.7427     | 0.4804 | 0.7694 | 0.8297 | 0.5953    | 0.5282 | 0.308  | 0.639  |
> > |Mixtral 4x7B| HC-SMoE   | **0.5729**  | **0.7285**     | **0.4573** | **0.7454** | **0.8018** | **0.5709** | **0.4571** | **0.27**   | **0.5523** |
> > |    | Fuzzy-Cmeans  | 0.5029  | 0.6654     | 0.3609 | 0.6456 | 0.7339 | 0.4704    | 0.3725 | 0.24   | 0.5343 |
> >
> > **Table r7.** Comparison of HC-SMoE and Fuzzy-Cmeans on Mixtral 8x7B-v0.1.

---

> ### Author Response · Authors · 2024-12-04
> **Rebuttal by Authors - Response to Comment 3**
>
> **C3.** The choice of the calibration dataset. I did not see any ablation study about the choice of the calibration dataset, and I think the performance of the proposed method should highly depend on the calibration dataset. If the calibration dataset is not comprehensive enough, e.g., not covering enough domain specific data, the clustering may not be very informative, which may lead to poor performance. For example, if we want the LLM to perform well on a law-related or medical-related task, can you also rely on the same calibration dataset used in the experiments?
>
> **Response:** We appreciate the reviewer for raising the question about the choice of the calibration dataset. To address this concern, we conducted two sets of experiments to demonstrate that the calibration dataset choice has minimal impact on the eight language tasks presented in the original manuscript and the additional experiments on the medical-related MedMCQA task [1].
>
> ***Calibration Dataset Comparison***
>
> We evaluated our approach using three different calibration datasets: C4 [2], MATH [3], and CodeQA [4]. These datasets vary in their domain focus:
>
> 1. C4: General-purpose and closely aligned with language tasks.
> 2. MATH: Focused on math question answering.
> 3. CodeQA: Addresses Python code question answering.
>
> The results presented in Table r8 demonstrate that the performance across the eight evaluated language tasks remains consistent, regardless of the calibration dataset used. Although C4 aligns most closely with general language tasks, the clustering results remain stable even when using domain-specific datasets such as MATH or CodeQA. This finding indicates that for general language tasks, the choice of calibration dataset has only a negligible impact on the effectiveness of our method.
>
> ***Domain-Specific Evaluation***
>
> To further evaluate domain-specific scenarios, we conducted experiments on MedMCQA, a challenging medical-related task. MedMCQA is a large-scale Multiple-Choice Question Answering (MCQA) dataset specifically designed to address real-world medical entrance exam questions. It contains over 194,000 high-quality multiple-choice questions sourced from the AIIMS and NEET PG entrance exams, covering 2,400 healthcare topics across 21 medical subjects. In the evaluations, a two-shot prompt is utilized as follows:
>
> “Please choose one option among A,B,C,D to answer the question.
>
> Question: Chronic urethral obstruction due to benign prismatic hyperplasia can lead to the following change in kidney parenchyma
>
> Options:
>
> A. Hyperplasia
>
> B. Hyperophy
>
> C. Atrophy
>
> D. Dyplasia
>
> Ans:C
>
> &nbsp;
>
> Question: All of the following are surgical options for morbid obesity except -
>
> Options:
>
> A. Adjustable gastric banding
>
> B. Biliopancreatic diversion
>
> C. Duodenal Switch
>
> D. Roux en Y Duodenal By pass
>
> Ans:D
>
> &nbsp;
>
> Question: {data.question}
>
> Options:
>
> A. {data.opa}
>
> B. {data.opb}
>
> C. {data.opc}
>
> D. {data.opd}
>
> Ans:”
>
> The model's output is assessed based on whether it generates {A, B, C, D} in the following three tokens.
>
> For these experiments, we used the validation set of MedMCQA for evaluation and its training set as the calibration dataset. Given that MedMCQA has an imbalanced answer distribution (950/735/617/514 for answers A/B/C/D) in the validation dataset, we provide a detailed evaluation using precision, recall, and F1-score metrics to ensure a comprehensive analysis.
>
> Our method demonstrated robust performance while maintaining its effectiveness even with datasets tailored to specialized domains. In particular, our approach outperformed all three baselines, which highlights its capability to handle tasks spanning diverse domains.
>
> We hope these additional experiments and analyses adequately address the reviewer's questions regarding the choice of the calibration dataset.
>
> [1] Ankit Pal *et al*., MedMCQA: A Large-scale Multi-Subject Multi-Choice Dataset for Medical domain Question Answering, Proceedings of the Conference on Health, Inference, and Learning, 2022
>
> [2] Colin Raffel *et al*., Exploring the Limits of Transfer Learning with a Unified Text-to-Text Transformer, Journal of machine learning research, 2020
>
> [3] Dan Hendrycks *et al*., Measuring mathematical problem solving with the math dataset, NeurIPS 2021.
>
> [4] Chenxiao Liu *et al*., CodeQA: A Question Answering Dataset for Source Code Comprehension, EMNLP 2021.

---

> > ### Author Response · Authors · 2024-12-04
> > **Rebuttal by Authors - Response to Comment 3 (cont.)**
> >
> > | Model           | Method        | Calib Dataset | Accuracy | Precision | Recall | F1    |
> > |-----------------|---------------|---------------|----------|-----------|--------|-------|
> > | Mixtral 8x7B    |               |               | 0.5930   | 0.5876    | 0.5918 | 0.5888|
> > | Mixtral 6x7B    | F-prune       | C4            | 0.3615   | 0.4109    | 0.3786 | 0.3498|
> > |     | S-prune       | C4            | 0.4794   | 0.4755    | 0.4721 | 0.4703|
> > |     | M-SMoE        | C4            | 0.1818   | 0.0909    | 0.1990 | 0.0634|
> > |     | HC-SMoE (ours) | C4            | 0.5018   | 0.4950    | 0.4785 | 0.4828|
> > |     | HC-SMoE (ours) | MATH          | **0.5110**   | **0.5044**    | **0.5034** | **0.5035**|
> > |     | HC-SMoE (ours) | Medmcqa       | 0.5004   | 0.4924    | 0.4835 | 0.4861|
> > |   Mixtral 4x7B | F-prune       | C4            | 0.3249   | 0.4363    | 0.3197 | 0.2404|
> > |  | S-prune       | C4            | 0.0000   | 0.0000    | 0.0000 | 0.0000|
> > |    | M-SMoE        | C4            | 0.0160   | 0.0720    | 0.0165 | 0.0263|
> > |  | HC-SMoE (ours) | C4            | 0.3817   | 0.4015    | 0.3883 | 0.3705|
> > |    | HC-SMoE (ours) | MATH          | **0.4268**   | **0.4139**    | **0.4060** | **0.4079** |
> > |    | HC-SMoE (ours) | Medmcqa       | 0.4130   | 0.3244    | 0.3064 | 0.3058|
> >
> > **Table r8.** Experimental results of HC-SMoE and three baselines on MedMCQA with Mixtral 8x7B and the compressed version of reducing experts to six and four per layer feeding with different calibration datasets.

---

> ### Author Response · Authors · 2024-12-04
> **Rebuttal by Authors - Response to Minor Issues (Comment 4, 5, 6, 7)**
>
> **C4.** In Fig. 1, why did you not compare the methods on the 14B model?
>
> **Response:** Fig. 1 presents a comparison of HC-SMoE with other baseline methods on Qwen1.5-MoE-A2.7B-Chat. The original Qwen1.5-MoE-A2.7B-Chat model comprises 14B parameters, which serves as the reference point denoted by a gray star to indicate its average accuracy across eight language tasks. The comparison excludes pruning methods for the 14B configuration since it represents the original, unmodified model architecture. The 11B and 8B model variants correspond to Qwen 45x2.7B and Qwen 30x2.7B respectively, which are derived through expert pruning at rates of 25% and 50%.
>
> The updated manuscript incorporates additional data points at 9.5B parameter count. These configurations result from expert pruning at rates of 37.5%, and provide more extensive comparative results.
>
> ---
>
> **C5.** Section 3.2.1 presents the method of similarity metric but contains a lot of discussions about related work.
>
> **Response:** We appreciate the reviewer's feedback. The rationale for incorporating related literature discussion in Section 3.2.1 stems from the necessity to establish context for the proposed similarity metrics through comparison with established methodologies. This contextualization reveals the foundations that motivated our design decisions and demonstrates the capabilities of the proposed approach to address specific limitations in existing methods. Moreover, we have refined the section to emphasize the mathematical formulations and core concepts that form the basis of the similarity metrics.
>
> We hope these revisions address the reviewer's question about the structure and content of Section 3.2.1.
>
> ---
> **C6.** In line 299/300, is alpha_i fixed or not? If it is fixed, will it also suffer from the issue that you mentioned in line 199-203 about frequency-based methods?
>
> **Response:** We appreciate the reviewer's question. In frequency-weighted merging, $\alpha_i$ is determined based on the usage frequency of the corresponding experts on the calibration dataset. We use frequency as a guide to weigh the experts that contribute to the merged expert.
>
> Regarding the question about lines 199-203, we had noted two key issues with M-SMoE's frequency-based approach for determining expert retention per layer. First, the number of experts per layer serves as a crucial hyperparameter for MoE, as it determines the capacity to store knowledge at that layer, and different tasks require varying amounts of knowledge across layers. HC-SMoE addresses this by decoupling the number of experts in each layer from frequency compared to M-SMoE. This ensures that this parameter remains independent of task-specific data.
>
> Moreover, in M-SMoE's merging strategy, experts with high usage frequency are never merged together due to an assumption that frequent activation indicates dissimilarity. However, high frequency alone does not guarantee that two experts produce dissimilar outputs. Our method addresses this through hierarchical clustering based on expert outputs. This enables accurate groupings of experts that exhibit similar behavior.
>
> The proposed frequency-weighted merging therefore avoids these two drawbacks. To demonstrate the robustness of the proposed method, we evaluated merging results across different datasets through validation experiments using average merging, as presented in Table 7 in our manuscript. The experimental results demonstrate that the choice of merging strategy produces minimal impact relative to the influence of the clustering algorithm and similarity metric.
>
> We hope the explanation clarifies the distinctions between our approach and M-SMoE's frequency-guided merging.
>
> ---
>
> **C7.** In Table 4, the best performance of 'ARC-c' should be the Average linkage using the Weight setting, right?
>
> **Response:** We appreciate the reviewer's attention to our typo. We have revised Table 4 in the updated manuscript to highlight the highest score of ARC-c in bold.

---

### Official Review · Reviewer_Pqdp · 2024-11-01

**Soundness:** 3
**Presentation:** 3
**Contribution:** 3
**Rating:** 5
**Confidence:** 4

**Summary:**

This work introduces Sparse Mixture-of-Experts (SMoE) models, which improve large language model performance without significantly increasing inference costs by activating only a subset of parameters. However, their high memory requirements hinder deployment. To address this, the authors propose Hierarchical Clustering for Sparsely activated Mixture of Experts (HC-SMoE), a task-agnostic framework that reduces SMoE parameters without retraining.

**Strengths:**

- HC-SMoE offers a practical solution for reducing parameters without the need for retraining, simplifying the implementation process.

- The task-agnostic nature of HC-SMoE allows for broader applicability across different language tasks, enhancing its versatility.

- The comprehensive experiments conducted on eight zero-shot language tasks provide strong empirical evidence of HC-SMoE's effectiveness in large-scale models like Qwen and Mixtral.

**Weaknesses:**

- While this work demonstrates competitive accuracy, it lacks a comprehensive assessment of efficiency metrics, such as speedup and memory usage. Given that efficiency is a key contribution, this aspect of the experimental results is essential.

- A theoretical analysis of the effectiveness of expert merging and HC-SMoE would enhance the understanding of the method's performance.

- Although HC-SMoE is validated on eight zero-shot language tasks, its effectiveness may vary in more complex tasks or domains, potentially limiting its broader applicability.

**Questions:**

Please refer to the Weakness.

---

> ### Author Response · Authors · 2024-12-04
> **Rebuttal by Authors - Response to Comment 1**
>
> **Rebuttal**: We appreciate the reviewer’s valuable feedback and effort spent on the review and would like to respond to the reviewer’s questions as follows.
>
> **C1.** While this work demonstrates competitive accuracy, it lacks a comprehensive assessment of efficiency metrics, such as speedup and memory usage. Given that efficiency is a key contribution, this aspect of the experimental results is essential.
>
> **Response**: We appreciate the reviewer for bringing attention to the importance of efficiency metrics in evaluating our proposed method. To address this, we have included an assessment of throughput (token per ms), latency (s), FLOPs, memory usage, and model size for both the Mixtral 8x7B and Qwen1.5-MoE-A2.7B-Chat models, as well as their pruned versions, in Table 10 of the updated manuscript. Furthermore, we have provided an additional description and discussion of these efficiency metrics in Appendix B.
>
> In the newly introduced experiments, we utilized the same calibration dataset as the main experiments, which consists of 32 sequences of 2,048 tokens sampled from the C4 corpus. As revealed in Table 10, HC-SMoE achieves significant reductions in memory usage and FLOPs by decreasing the number of experts while maintaining the same throughput and latency. Although reducing the number of experts may inherently decrease router latency by reducing output channels, we have chosen to retain the original router weights to avoid compromising accuracy. As a result, the router operates as if the original number of experts is preserved, while experts within the same group produce identical outputs through the corresponding merged expert.
>
> We hope that this evaluation clarifies the computational and memory costs of the proposed model. Below is the table we provided in the manuscript's Table 10.
>
> | Models            | Throughput          | Latency            | GFLOPs | Memory   | Model Size |
> |-------------------|----------------------|--------------------|--------|----------|------------|
> | Mixtral 8x7B      | 13.45 ± 1.30         | 2.854 ± 0.333      | 2989   | 87.49GB  | 46.7B      |
> | Mixtral 6x7B      | 13.87 ± 0.47         | 2.666 ± 0.093      | 2267   | 66.49GB  | 35.4B      |
> | Mixtral 4x7B      | 13.96 ± 0.65         | 2.599 ± 0.166      | 1546   | 45.49GB  | 24.2B      |
> |-------------------|----------------------|--------------------|--------|----------|------------|
> | Qwen 60x2.7B      | 24.08 ± 0.17         | 1.593 ± 0.168      | 916    | 27.04GB  | 14.3B      |
> | Qwen 45x2.7B      | 23.95 ± 0.24         | 1.541 ± 0.011      | 717    | 21.23GB  | 11.2B      |
> | Qwen 30x2.7B      | 23.16 ± 0.42         | 1.583 ± 0.034      | 518    | 15.44GB  | 8.1B       |
>
> **Table 10 (in our manuscript).**  Evaluation of computational and memory efficiency across multiple models. For Mixtral: Mixtral 8x7B (original), Mixtral 6x7B (25% pruned), and Mixtral 4x7B (50% pruned). For
> Qwen1.5-MoE-A2.7B-Chat: Qwen 60x2.7B (original), Qwen 45x2.7B (25% pruned), and Qwen
> 30x2.7B (50% pruned). All measurements use identical input sequences and include throughput
> (tokens per ms), latency (s), GFLOPs, model memory, and model size (number of parameters).

---

> ### Author Response · Authors · 2024-12-04
> **Rebuttal by Authors - Response to Comment 2**
>
> **C2.** A theoretical analysis of the effectiveness of expert merging and HC-SMoE would enhance the understanding of the method's performance.
>
> **Response:** We are thankful to the reviewer for paying attention to the theoretical analysis of HC-SMoE and expert merging. Our response addresses the theoretical foundations and benefits of expert merging in our approach.
>
> The effectiveness of expert merging originates from fundamental principles in neural network optimization and representation learning. Multiple experts that exhibit similar output patterns likely encode redundant information or perform comparable transformations on the input data. The identification and merging of these functionally similar experts enables model redundancy reduction while it preserves essential computational capabilities.
>
> The weighted merging process is preferred since it preserves model performance through two fundamental mechanisms. First, experts with similar outputs operate within the same functional subspace, which indicates their capture of related features and transformations. The merged expert retains this shared functionality through the elimination of redundant parameters. Second, the weighted merging approach maintains the relative order of row vectors in the expert network weights, which ensures minimal perturbation to the learned feature representations. Compared to weighted merging, fix-dominant merging, the third merging method we proposed, disturbs the feature order inside the weight matrices of the expert network, please refer to appendix A.2 for detailed explanation.
>
> The experimental evidence in Table 7 of our manuscript demonstrates that effective clustering serves as the primary determinant of model performance. This finding emerges from the observation that even basic average merging strategies achieve superior results. The theoretical foundation for this phenomenon rests in the functional equivalence among experts within each cluster, which manifests through their consistent output patterns. This fundamental property ensures robust model performance across different merging methodologies.

---

> ### Author Response · Authors · 2024-12-04
> **Rebuttal by Authors - Response to Comment 3**
>
> **C3**. Although HC-SMoE is validated on eight zero-shot language tasks, its effectiveness may vary in more complex tasks or domains, potentially limiting its broader applicability.
>
> **Response:** To evaluate HC-SMoE on more complex domain-specific tasks, we conducted additional experiments on MedMCQA [1], a challenging medical question-answering dataset. Table r4 presents the experimental results, which demonstrate HC-SMoE's robust performance in domain-specific applications. The performance of HC-SMoE exceeds all three baseline methods in this specialized domain.
>
> For the reviewer's reference, MedMCQA comprises a large-scale Multiple-Choice Question Answering (MCQA) dataset designed for real-world medical entrance examinations. The dataset contains over 194,000 high-quality multiple-choice questions from AIIMS and NEET PG entrance exams, which span 2,400 healthcare topics across 21 medical subjects. Our experiments utilize a two-shot prompt format:
>
> “Please choose one option among A,B,C,D to answer the question.
>
> Question: Chronic urethral obstruction due to benign prismatic hyperplasia can lead to the following change in kidney parenchyma
>
> Options:
>
> A. Hyperplasia
>
> B. Hyperophy
>
> C. Atrophy
>
> D. Dyplasia
>
> Ans:C
>
> &nbsp;
>
> Question: All of the following are surgical options for morbid obesity except -
>
> Options:
>
> A. Adjustable gastric banding
>
> B. Biliopancreatic diversion
>
> C. Duodenal Switch
>
> D. Roux en Y Duodenal By pass
>
> Ans:D
>
> &nbsp;
>
> Question: {data.question}
>
> Options:
>
> A. {data.opa}
>
> B. {data.opb}
>
> C. {data.opc}
>
> D. {data.opd}
>
> Ans:”
>
> The evaluation protocol assesses whether the model outputs {A, B, C, D} in the subsequent three tokens.
>
> The experimental methodology employs the MedMCQA validation set for evaluation and its training set for calibration. Due to the imbalanced answer distribution in the validation dataset (950/735/617/514 for answers A/B/C/D), we present a comprehensive analysis through precision, recall, and F1-score metrics.
>
> | Model        | Method           | Accuracy | Precision | Recall | F1    |
> |--------------|------------------|----------|-----------|--------|-------|
> | Mixtral 8x7B |                  | 0.5930   | 0.5876    | 0.5918 | 0.5888 |
> | Mixtral 6x7B | F-prune          | 0.3615   | 0.4109    | 0.3786 | 0.3498 |
> |              | S-prune          | 0.4794   | 0.4755    | 0.4721 | 0.4703 |
> |              | M-SMoE           | 0.1818   | 0.0909    | 0.1990 | 0.0634 |
> |  |    HC-SMoE (ours)     | **0.5018**   | **0.4950**    | **0.4785** | **0.4828** |
> | Mixtral 4x7B | F-prune          | 0.3249   | 0.4363    | 0.3197 | 0.2404 |
> |              | S-prune          | 0.0000   | 0.0000    | 0.0000 | 0.0000 |
> |              | M-SMoE           | 0.0160   | 0.0720    | 0.0165 | 0.0263 |
> |  |     HC-SMoE (ours)   | **0.3817**   | **0.4015**    | **0.3883** | **0.3705** |
>
> **Table r4.** Experimental results of HC-SMoE and three baselines on MedMCQA with Mixtral 8x7B and the compressed version of reducing experts to six and four per layer.
>
> [1] Ankit Pal *et al*., MedMCQA: A Large-scale Multi-Subject Multi-Choice Dataset for Medical domain Question Answering, Proceedings of the Conference on Health, Inference, and Learning, 2022

---

### Official Review · Reviewer_KGGX · 2024-11-04

**Soundness:** 3
**Presentation:** 3
**Contribution:** 3
**Rating:** 6
**Confidence:** 3

**Summary:**

The paper presents a retraining free experts merging approach which employs a hierarchical clustering strategy. The authors claim that using expert outputs as the similarity metric for clustering is more effective compared with using router logins or weights employed by prior works. The experimental results reveal the proposed approach gains more performance improvements compared with existing methods across various benchmarks.

**Strengths:**

1. An output based similarity metric of expert clustering is proposed, which is more effective than previous works.
2.The experimental results compared with the previous methods are very good.

**Weaknesses:**

There is a lack of theoretical analysis on the performance of expert clustering using different similarity metrics.

**Questions:**

I am wondering the results or analysis for more extreme expert reduction scenarios, such as reducing to 25% or 10% of the original experts. This would give insight into how the method performs under more aggressive compression.

---

> ### Author Response · Authors · 2024-12-04
> **Rebuttal by Authors - Response to Comment 1**
>
> **Rebuttal**:
>
> We appreciate the reviewer’s valuable feedback and effort spent on the review and would like to respond to the reviewer’s questions as follows.
>
> **C1.** There is a lack of theoretical analysis on the performance of expert clustering using different similarity metrics.
>
> **Response:**
> Our response addresses this through three parts: formal metric analysis, empirical evaluation, and experimental results.
>
> ***Formal Analysis of Similarity Metrics***
>
> We begin by defining three similarity metrics evaluated in our study. For an input token $x$:
> - Router logits $R(x)$: Calculated as $R(x) = x \cdot W_R$, where $W_R$ represents the router weight.
> - Expert weights $W$: Concatenated expert weights, $W = \text{flatten}(W_{gate}||W_{down}||W_{up})$, which remain independent of the input $x$.
> - Expert output $E(x)$: Defined as the output of each expert without multiplying routing scores, expressed as $E(x) = (\sigma(x\cdot W_{gate}) \odot (x\cdot W_{up}))\cdot W_{down}$.
>
> Router logits primarily reflect the assignment preference of tokens and depend on input distributions. These logits capture local routing decisions rather than global functional similarity between experts in an MoE layer. Similar router logits may indicate shared importance for certain tokens but fail to reflect functional equivalence.
>
> Expert weights present two limitations. First, their processing requires substantial computational overhead. Second, the high dimensionality of weight spaces relative to input/output dimensions obscures functional relationships. Different weight configurations can produce similar outputs, and similar weights may generate functionally distinct behaviors due to the complex dependencies in high-dimensional parameter spaces.
>
> Expert outputs directly reflect the functional behavior of experts in an MoE layer [1-3]. The average expert outputs over a calibration dataset with $T$ tokens, computed as $\frac{1}{T}\sum^T_{i=1}E(x_i)$, capture this functional behavior. This establishes expert output as the most appropriate similarity metric for clustering.
>
> ***Experimental Validation***
>
> Our analysis of layer outputs uses two quantitative criteria:
> 1. L2 distance: $|| T(x) - S(x) ||_2$​, where $T(x)$ and $S(x)$ represent the outputs of the original and pruned models. Lower values indicate better performance.
> 2. Cosine similarity: $\text{cosine-similarity}(T(x), S(x))$. Higher values indicate better alignment.
>
> Table r1 shows that hierarchical clustering based on expert outputs achieves the lowest L2 error and highest cosine similarity with original model outputs across 25% and 50% pruning ratios. As demonstrated in the literature [1,2,3], outputs that closely resemble the original model correlate strongly with improved performance in reduced model configurations. These results validate that experts with similar outputs maintain similar functionality and can serve as strong candidates for merging with minimal performance impact.
>
> | Pruned Ratio | Cluster       | Metric          | L2 Error | Cosine Similarity |
> |--------------|---------------|---------------------|---------------------|-------------------|
> | **25% (45e)**    | Hierarchical  | *expert-output*   | **3806.8332**          | **0.9972**            |
> |                          | Hierarchical  | *weight*          | 9307.8344          | 0.9874            |
> |                          | Hierarchical  | *router-logits*   | 7572.2246          | 0.9897            |
> | **50% (30e)**    | Hierarchical  | *expert-output*  | **8142.6961**          | **0.9878**            |
> |                           | Hierarchical  | *weight*          | 11400.2119         | 0.9791            |
> |                            | Hierarchical  | *router-logits*  | 10022.4158         | 0.9826            |
>
> **Table r1**. The L2 error and cosine similarity of last layer output between the original model Qwen1.5-MoE-A2.7B-Chat and the model produced by hierarchical clustering with three different metrics at 25% and 50% pruning ratio.
>
>
> ***Zero-Shot Evaluation Results***
>
> The zero-shot evaluation results on eight language tasks (Tables 4 and 6 in the manuscript) demonstrate that output-based clustering achieves higher average accuracy compared to weights or router logits across clustering methods. This improved performance follows from the direct relationship between expert outputs and model functionality, which preserves essential computational patterns after merging.
>
> We thank the reviewer for the question and hope this analysis provides sufficient clarification.
>
>
> [1] Liang *et al*., Less is More: Task-aware Layer-wise Distillation for Language Model Compression, ICML 2023
>
> [2] Romero *et al*., FitNets: Hints for Thin Deep Nets, ICLR 2015
>
> [3] Hinton *et al*., Distilling the Knowledge in a Neural Network, NeurIPS 2014

---

> ### Author Response · Authors · 2024-12-04
> **Rebuttal by Authors - Response to Comment 2**
>
> **C2.** I am wondering about the results or analysis for more extreme expert reduction scenarios, such as reducing to 25% or 10% of the original experts. This would give insight into how the method performs under more aggressive compression.
>
> **Response:** We appreciate the reviewer for raising this question. To evaluate extreme pruning scenarios, we conducted additional experiments at substantial compression rates of 62.5% and 75%. Our analysis compares HC-SMoE against four established baselines: *F-prune*, *S-prune*, *O-prune* and *M-SMoE*.  The results for the Qwen1.5-MoE-A2.7B-Chat and Mixtral 8x7B models are presented in Table r2 and r3, respectively.
>
> Several benchmark tasks employ multiple-choice formats. ARC-c, ARC-e, HellaSwag, MMLU, and OBQA require selection from four options, with 0.25 as the random-guess baseline. BoolQ, RTE, and Winogrande utilize binary choices, with 0.5 as the random-guess baseline. Scores in proximity to these baselines indicate a substantial deterioration of model capabilities.
>
> Under extreme reduction settings, our experiments reveal that all baselines experience significant accuracy degradation, with performance often falling below random-guess baselines. In contrast, HC-SMoE maintains competitive accuracy even at a 75% reduction through its output-based clustering and merging strategy. This finding can be attributed to the fact that experts with similar outputs are likely to capture related features or patterns in the data. Merging these experts allows the model to preserve essential information while reducing redundancy, which enables a more compact representation without compromising performance.
>
> The evaluation excludes 90% reduction scenarios because such extreme pruning would leave only one expert in the Mixtral model, which conflicts with its top-2 routing policy. Nevertheless, the results obtained at 62.5% and 75% pruning rates provide substantial evidence that HC-SMoE can effectively operate under extreme compression settings.
>
> | Model            | Method    | ARC-c  | ARC-e  | BoolQ  | HellaSwag | MMLU   | OBQA   | RTE    | Winogrande | Average |
> |------------------|-----------|--------|--------|--------|-----------|--------|--------|--------|------------|---------|
> | Qwen 60x2.7B     |           | 0.3951 | 0.7012 | 0.8135 | 0.5932    | 0.6047 | 0.31   | 0.7329 | 0.6559     | 0.6008  |
> | Qwen 23x2.7B     | F-prune   | 0.2287 | 0.3763 | 0.5957 | 0.3627    | 0.2413 | 0.186  | 0.5668 | 0.528      | 0.3857  |
> |                 | S-prune   | 0.215  | 0.42   | 0.5945 | 0.3307    | 0.2725 | 0.166  | 0.5343 | 0.5406     | 0.3842  |
> |                 | MC-SMoE   | 0.2014 | 0.2803 | 0.441  | 0.2743    | 0.2292 | 0.158  | 0.4982 | 0.5114     | 0.3242  |
> |                 | HC-SMoE (ours)  | **0.3319** | **0.5720**  | **0.7554** | **0.4111**    | **0.3957** | **0.216**  | **0.6606** | **0.6117**     | **0.4943**  |
> | Qwen 15x2.7B     | F-prune   | 0.2176 | 0.3026 | 0.5269 | 0.2871    | 0.2358 | 0.154  | 0.5018 | 0.5185     | 0.343   |
> |                 | S-prune   | 0.1954 | 0.3114 | 0.5275 | 0.2803    | 0.2537 | 0.136  | 0.5199 | 0.5122     | 0.3421  |
> |                 | MC-SMoE   | 0.1903 | 0.3035 | 0.3966 | 0.2741    | 0.2295 | 0.16   | 0.5199 | 0.5028     | 0.3221  |
> |                 | HC-SMoE (ours)   | **0.2662** | **0.5034** | **0.7046** | **0.3664**    | **0.3629** | **0.196**  | **0.6173** | **0.5777**     | **0.4493**  |
>
> **Table r2.** Zero-shot performance evaluation of HC-SMoE and three baseline methods on Qwen1.5-MoE-A2.7B-Chat with expert reduction to 23 and 15 per layer. We exclude O-prune for this experiment due to the large search space.

---

> ### Author Response · Authors · 2024-12-04
> **Rebuttal by Authors - Response to Comment 2 (cont.)**
>
> | Model          | Method    | ARC-c   | ARC-e   | BoolQ   | HellaSwag | MMLU    | OBQA    | RTE     | Winogrande | Average | - | **Time (s)** |
> |----------------|-----------|---------|---------|---------|-----------|---------|---------|---------|------------|---------|------------------|-----------------|
> | Mixtral 8x7B   |           | 0.5648  | 0.8422  | 0.8505  | 0.649     | 0.6712  | 0.35    | 0.7112  | 0.7593     | 0.6748  |       -           |                 |
> | Mixtral 3x7B   | F-prune   | 0.2253  | 0.399   | 0.6024  | 0.3663    | 0.2414  | 0.168   | 0.5379  | 0.5249     | 0.3832  |          -        | 61.070          |
> |                | S-prune   | 0.2082  | 0.3826  | 0.5951  | 0.3648    | 0.2315  | 0.154   | 0.509   | 0.5383     | 0.3729  |          -        | 54.000          |
> |                | O-prune   | **0.4471**  | **0.7210**  | **0.7761**  | 0.5377    | 0.3847  | 0.264   | **0.5921**  | **0.7024**     | **0.5531**  |        -          | 2530.584        |
> |                | MC-SMoE   | 0.2125  | 0.2963  | 0.6131  | 0.2699    | 0.2513  | 0.126   | 0.5162  | 0.5185     | 0.3505  |     -             | 43.544          |
> |                | HC-SMoE (ours) | 0.4078 | 0.7138 | 0.7755 | **0.5402** | **0.4156** | **0.268** | 0.5451 | 0.7001 | 0.5458 |     -             | 253.299     |
> | Mixtral 2x7B   | F-prune   | 0.2329  | 0.2689  | 0.6214  | 0.2681    | 0.2574  | 0.15    | 0.491   | 0.5162     | 0.3507  |       -           | 61.868          |
> |                | S-prune   | 0.2022  | 0.2929  | 0.6193  | 0.2942    | 0.2356  | 0.142   | 0.5199  | 0.5083     | 0.3518  |      -            | 55.718          |
> |                | O-prune   | 0.3481  | 0.654   | 0.7043  | **0.4846**    | 0.3163  | 0.214   | **0.5451**  | **0.6685**     | 0.4919  |         -         | 1181.15         |
> |                | MC-SMoE   | 0.2116  | 0.2908  | 0.6196  | 0.2697    | 0.237   | 0.132   | 0.4729  | 0.5107     | 0.343   |         -         | 43.778          |
> |                | HC-SMoE (ours) | **0.3746** | **0.6721** | **0.7541** | 0.4786 | **0.3606** | **0.236** | 0.5307 | 0.6582 | **0.5081** |    -              | 267.134     |
>
> **Table r3.** Zero-shot performance evaluation of HC-SMoE and four baseline methods on Mixtral 8x7B-v0.1 with expert reduction to three and two per layer. The runtime for each algorithm is provided in seconds. Please note that the computational requirements of O-prune exceed other methods by a factor of five to ten, which presents significant practical limitations for model compression tasks.

---

### Author Response · Authors · 2024-12-04
**Author Rebuttal by Authors**

**Rebuttal**:

We appreciate the area chairs' time and effort as well as the reviewers' feedback. We have diligently responded to each reviewer's questions in the individual comments to address their specific concerns. Furthermore, we have updated the manuscript according to the suggestions from the reviews.

**The updated manuscript includes the following modifications in red:**
- **Fig. 1**: Additional results include pruning 37.5% of the experts in Qwen1.5-MoE-A2.7B-Chat. This addition enables a smoother evaluation of HC-SMoE and baseline methods across different pruning ratios.
- **Section 3.1**: Enhanced formalization of the clustering and merging expert framework. We introduce $C$ to represent a cluster and $r$ to denote the average number of experts per layer after merging.
- **Section 3.2.1**: The section extends the mathematical formalization of expert weights and router logits as similarity metrics.
- **Section 3.2.2**: The section establishes the mathematical foundations for expert distance calculation in Eq. (4) and derives the expressions for three linkage methods in Eqs. (5, 6, and 7). This formulation establishes the theoretical framework of HC-SMoE's clustering methodology.
- **Section 3.2.3**: The updated expert merging equation aligns with the new variable definitions introduced in Section 3.1.
- **Appendix B**: The appendix incorporates a discussion of computational and memory costs for the Mixtral 8x7B and Qwen1.5-MoE-A2.7B-Chat models in both original and merged versions. Table 10 presents detailed evaluations of throughput, latency, GFLOPs, memory usage, and model size.

---

### Note · Authors · 2025-01-23

I have read and agree with the venue's withdrawal policy on behalf of myself and my co-authors.